# Compounds from plantar foot sweat, nesting material, and urine show strain patterns associated with agonistic and affiliative behaviors in group housed male mice, *Mus musculus*

Amanda J. Barabas●[1]*, Helena A. Soini[2], Milos V. Novotny[2], David R. Williams[2], Jacob A. Desmond[2], Jeffrey R. Lucas[3], Marisa A. Erasmus[1], Heng-Wei Cheng[4], Brianna N. Gaskill[1]

1 Department of Animal Science, Purdue University, West Lafayette, Indiana, United States of America,
2 Department of Chemistry and Institute for Pheromone Research, Indiana University, Bloomington, Indiana, United States of America, 3 Department of Biological Science, Purdue University, West Lafayette, Indiana, United States of America, 4 USDA-ARS, Livestock Behavior Research Unit, Purdue University, West Lafayette, Indiana, United States of America

* abarabas@purdue.edu

## Abstract

Excessive home cage aggression often results in severe injury and subsequent premature euthanasia of male laboratory mice. Aggression can be reduced by transferring used nesting material during cage cleaning, which is thought to contain aggression appeasing odors from the plantar sweat glands. However, neither the composition of plantar sweat nor the deposits on used nesting material have been evaluated. The aims of this study were to (1) identify and quantify volatile compounds deposited in the nest site and (2) determine if nest and sweat compounds correlate with social behavior. Home cage aggression and affiliative behavior were evaluated in 3 strains: SJL, C57BL/6N, and A/J. Individual social rank was assessed via the tube test, because ranking may influence compound levels. Sweat and urine from the dominant and subordinate mouse in each cage, plus cage level nest samples were analyzed for volatile compound content using gas chromatography-mass spectrometry. Behavior data and odors from the nest, sweat, and urine were statistically analyzed with separate principal component analyses (PCA). Significant components, from each sample analysis, and strain were run in mixed models to test if odors were associated with behavior. Aggressive and affiliative behaviors were primarily impacted by strain. However, compound PCs were also impacted by strain, showing that strain accounts for any relationship between odors and behavior. C57BL/6N cages displayed the most allo-grooming behavior and had high scores on sweat PC1. SJL cages displayed the most aggression, with high scores on urine PC2 and low scores on nest PC1. These data show that certain compounds in nesting material, urine, and sweat display strain specific patterns which match strain specific behavior patterns. These results provide preliminary information about the connection between home cage

**Data Availability Statement:** All relevant data are within the manuscript and its Supporting Information files.

**Funding:** This study was funded by a grant from the Purdue Center for Animal Welfare Science, awarded to B.N.G, J.R.L, M.A.E, and H.W.C. The funders had no role in study design, data collection and analysis, decision to publish, or preparation of the manuscript. Further, 3,5-diethyl- 2-hydroxycyclopent-2-en-1-one structure was verified with funds from the National Science Foundation grant CHE1726633.

**Competing interests:** The authors have declared that no competing interests exist.

compounds and behavior. Salient compounds will be candidates for future controlled studies to determine their direct effect on mouse social behavior.

## Introduction

Aggression among group housed male mice is one of the most common reasons for premature euthanasia and reduces preclinical research data validity and reproducibility [1–3]. Individual housing appears to be a simple solution, but it comes with its own welfare concerns [4]. Mice form complex social structures in the wild [5, 6], which is why group housing for laboratory mice is recommended [7]. Enrichment is commonly suggested to reduce home cage aggression, but results are often inconsistent [2]. Nonetheless, nesting material is one of the most reliable and recommended types of enrichment, particularly for reducing aggression after cage cleaning [2, 8]. Routine cage cleaning is a known trigger of escalated aggression in males [9] with time periods of social unrest peaking approximately 15 to 45 minutes afterward [10, 11]. However, this aggression is reduced when a portion of the existing nest is transferred to the new cage [12]. Accordingly, nest transfer has become a widely used practice, but there is no empirical evidence to explain how it decreases aggression.

Although there are minimal data, the prevalent theory explaining these effects focuses on scent cue preservation. The familiar odors within the nesting material may include pheromones, which are commonly produced as volatile organic compounds (VOCs) and play a prominent role in regulating mammalian social interactions [13]. While pheromones are the most recognized odor signal, odors must meet strict criteria to be considered a pheromone: physiologically relevant concentrations must produce reliable effects in a bioassay [14, 15]. Individualized scent profiles can also relay information, and mice rely heavily on both pheromones and individual scent cue mixtures for communication and conspecific recognition [15–18]. The disruption of these scent cues can in turn lead to aggressive interactions [19].

While odor signals relay a variety of messages, most of the literature on male, intra-sex, signaling focuses on urine borne signals that are connected to territory marking in wild mice and ultimately promote aggression in the laboratory [6, 20–27]. In contrast, little is known about odor signals that may reduce aggression or promote affiliative behaviors among male mice. In pigs, synthetic androstenone and maternal mammary pheromones effectively reduce aggression in newly mixed groups of prepubescents [28, 29], but, to the best of our knowledge, compounds with similar effects in mice have not been identified. Affiliative behaviors, for example, are performed to strengthen social bonds between conspecifics, and examples in mice include allo-grooming and group sleep [21]. While aggression and affiliative behavior patterns do not always oppose each other [30], it has been proposed that they can be different context dependent strategies used for resource control. Affiliative behaviors are deemed more beneficial when resources are abundant, such as in a captive enclosure with free food and water access [31]. However, almost all work on domestic murine social behavior focuses on encounters with unfamiliar mice in a testing arena. Affiliative patterns between adult males in the home cage have been largely unexplored and will be examined here.

Despite the lack of explicit evidence, it has been suggested that nesting material contains an aggression appeasing odor signal [12]. Specifically, the nest site appears to act as a depository for secretions from the plantar sweat glands which are believed to appease aggression [8, 12]. However, there is little empirical data describing the properties of plantar sweat. Laboratory mice only have one type of sweat gland, eccrine glands, which are found on their food pads

[5]. These glands produce an oily substance that is associated with maintaining traction during mobility, marking territory boundaries, and colony member recognition [21, 32, 33]. However, the only study to specifically link plantar sweat to a behavioral response demonstrated that the presence of sweat increases locomotion in stranger mice [34].

To date, there are no published studies that explore the mechanism behind the reduction in aggression observed in response to used nest material or whether odors exist that can promote affiliative behaviors in mice. Providing nesting material is becoming standard practice for laboratory mice and its transfer during cage cleaning helps reduce aggression although it does not completely eliminate it. In order to understand what in the nest is specifically effective at altering mouse behavior, we must have better insight into the chemical signals deposited there and where they come from. Once these specific signals have been identified, further research can examine methods to develop compounds that could then be added to mice environments to help reduce aggression. Additionally, there are no reports that quantitatively analyze the VOC contents of murine plantar sweat, which has historically been suggested as the source of nesting material odor deposits. Therefore, the first aim of this experiment was to quantify compounds deposited within the nests of mouse strains known to exhibit different aggression levels and link them to plausible sources. Our working hypothesis was that the compounds present on the nests would exhibit strain specific properties. We predicted that chemical analyses of the nests from historically peaceful mice would contain VOCs in different proportions than those from the nests of historically aggressive males; in particular, they would contain higher levels of VOCs originating in plantar sweat and lower levels of VOCs originating in urine. To do this, we used three strains known for varying aggression levels: SJL (high aggression), C57BL/6 (moderate aggression), and AJ (low aggression). Our second aim was to determine whether these VOC profiles are related to mouse social behavior. Our working hypothesis was that VOC profiles from the nest and sweat correlate with social behavior in group housed males, with the assumption that behavior is affected similarly across strains. We predicted that these odor profiles would be associated with lower rates of aggressive behavior and/or higher rates of affiliative behavior. In contrast, profiles from urine would be associated with higher rates of aggression. Social behavior was taken as a cage level measure, while odor profiles were taken from individuals based on dominance rank in the tube test [35].

This study served as the first step in a series of projects that aim to identify and validate whether the VOCs identified are true murine pheromones, based on criteria summarized by Wyatt [14, 15]. The goal of the current study was solely to compare profiles across experimental groups and identify molecules that align with quantified behavioral measures.

## Results

Cages containing five male mice of SJL/JOrlIcoCrl (SJL), C57BL/6NCrl (B6), or A/JCr (AJ) strain were kept for one week (n = 8 cages per strain; N = 24 total cages). At the end of the week, samples of used nesting material were taken from each cage. Samples of sweat, saliva, and urine were also collected from each cage's dominant and subordinate mouse as determined by the tube test. All samples were analyzed using gas chromatography- mass spectrometry (GC-MS) and proportions of each sample's VOCs were evaluated. However, saliva samples were only sufficient enough for qualitative assessments. One AJ nest sample was excluded from analyses due to a flooded cage during the study (leaving N = 23); two sweat samples (one B6 and one AJ) were excluded due to missing labels (leaving N = 46); and six SJL mice did not urinate when stimulated (leaving N = 42). See Methods for further details.

Video data from days 1, 2, and 7 were collected and analyzed for social interactions (mediated aggression, escalated aggression, social investigation, allo-grooming, and group sleep)

and nesting behaviors (paw nesting and oral nesting). Full behavioral descriptions can be found in the methods. Ultimately, we calculated the proportion of time that each behavior was observed. Unless otherwise indicated, behavior proportions represent values for all three days observed. An overview of the sample size used in each analysis is provided (S1 Table).

## Sample VOC profiles

To address aim 1, we identified or tentatively identified 32 compounds across all sample types (Table 1). Among those, 53% were found in at least 2 sample types; 6% were unique to nest samples; 22% were unique to sweat; 16% were unique to saliva; and 3% were unique to urine (Fig 1). Subsequent analyses excluded saliva samples due to low sample volumes (see Gas Chromatography- Mass Spectrometry Analysis in Methods). As indicated in Table 1, nesting material and urine samples shared many previously identified mouse urinary compounds. In turn, sweat samples showed several cyclic ketone compounds also found in the nesting material, which were not detected in urine samples.

## Strain and VOC profiles

Visual examination of sample profiles using non-metric multidimensional scaling (NMDS) showed sample separation based on strain for nesting material, sweat, and urine VOC profiles (Fig 2). Social (dominant versus subordinate) ranking effects were not distinguishable in the sweat and urine samples (Fig 2B and 2C).

Analyses using the Adonis test showed that strain significantly impacted VOC proportions in nesting material, sweat, and urine (p values<0.01; S2 Table). Social rank did not significantly influence VOC proportions in sweat or urine (p values> 0.05; S2 Table).

Additionally, proportions of two urinary pheromones (β-farnesene and 2-sec-butyl-thiazoline (SBT)) were analyzed based on strain and social rank. Higher quantities of both pheromones have been reported in dominant compared to subordinate urine [36], so we used Restricted Maximum Likelihood mixed models to confirm rankings from the tube test. Here, proportions of neither of these pheromones differed by social rank (p values > 0.05; S3 Table), although AJ and B6 mice produced more SBT than SJL mice (Tukey: p< 0.05). Even though we assigned a "dominant" or "subordinate" label to the sampled mice, dominance rank was based solely on the tube test, and may not reflect in-cage behavior. Since dominance rank was not a significant source of variation between sweat and urine samples, the two samples from each cage were averaged together to give single cage mean values for subsequent analyses. However, in cases where only one sample was collected from a cage (see Methods for additional information), that sample alone was used for analysis (S1 Table).

## VOC profiles and social interactions

Separate principal component analyses (PCA) were run for each sample type and the behavior data. Strong PC loadings (absolute value ≥ 0.300) considered important are indicated in gray highlighted cells and bold black numbers in Table 2. Influential PCA components from each data set were kept for mixed models. Aggressive behaviors and social investigation had high positive loadings on PC1 while allo-grooming had a strong negative loading. On PC2, group sleep and allo-grooming behaviors had high loadings. Loading values for all influential sweat, nest, urine, and behavior PCs are listed in Table 2.

To address aim 2, two mixed models were run and p values were corrected using the sequential Bonferroni procedure [37]: one for each behavior PC. All significant VOC PCs and strain were included as independent variables, as well as two covariate measures: average cage nest complexity score, and dominance linearity as measured by Landau's H. Please refer to the

**Table 1. List of identified compounds across sample type in order of ascending run time.**

| Compound | SIC m/z | Nesting Material Rt (min) | Sweat Rt (min) | Urine Rt (min) | Saliva Rt (min) |
|---|---|---|---|---|---|
| acetic acid[1] | 60 | 3.44 | 3.44 | 3.44 | 3.44 |
| 5,5-dimethyl-2-ethyl-4,5-dihydrofuran[2, #] | 126 | 5.56 | | 5.56 | |
| 2-furanmethanol[1] | 98 | 7.83 | 7.83 | 7.83 | 8.05 |
| Z-5,5-dimethyl-2-ethylidenetetrahydrofuran[2, #] | 126 | 7.98 | | 7.98 | |
| E-5,5-dimethyl-2-ethylidenetetrahydrofuran[2, #] | 126 | 9.38 | | 9.38 | |
| *1,2-cyclopentadione | 98 | 10.85 | | 10.82 | 11.08 |
| 6-hydroxy-6-methyl-3-heptanone[2, #] | 127 | 11.47 | | 11.47 | |
| 3-methyl-(2(H)-furanone[1] | 98 | | 13.03 | | |
| **a ketone (m/z 55, 84, 114) | 114 | | | | 14.87 |
| 2-isopropylthiazoline[2, #] | 114 | 15.57 | | 15.57 | |
| methylcyclopentenolone[1] | 112 | 16.1 | 16.1 | | |
| limonene[1] | 68 | | | | 16.21 |
| *2-hydroxybenzaldehyde | 122 | 17.17 | | | |
| dehydrobrevicomin[2, #] | 111 | 17.6 | | 17.6 | |
| 3,4-dimethyl-1,2-cyclopentadione[1] | 111 | | 19.15 | | |
| o-toluidine[1] | 107 | | 19.86 | 19.18 | |
| 2-sec-butylthiazoline[2, #] | 115 | 21.2 | | 21.2 | 21.16 |
| nonanal[1] | 98 | | | | 21.52 |
| 3-ethyl-2-hydroxy-2-cyclopenten-1-one (ethylcyclopentenolone)[1] | 126 | | 22.02 | | |
| *n-formylmorpholine | 115 | | 22.66 | | |
| *5-ethylthiazolidine | 117 | 26.64 | 24.65 | 24.64 | 24.75 |
| 3,5-diethyl- 2-hydroxycyclopent-2-en-1-one[2] | 126 | | 29.69 | | |
| indole[1] | 117 | | 33.82 | | |
| **m/z 126 compound | 111 | | | | 34.28 |
| **m/z 152 compound | 70 | 41.38 | | | |
| geranylacetone[1] | 69 | 43.93 | 43.92 | | 43.92 |
| β-farnesene[2, #] | 69 | 44.26 | | 44.26 | |
| α-farnesene[2, #] | 69 | | | 50.65 | |
| methyldihydrojasmonate[1] | 69 | | | | 55.54 |
| hexadecanol[1] | 55 | | 67.92 | 67.92 | |
| hexadecanoic acid[1] | 60 | 72.12 | 72.12 | 72.12 | |
| octadecanoic acid[1] | 60 | 76.41 | 76.41 | 76.41 | |

[1] Identified using commercial standard compounds.

[2] Identified using in-house spectrum database based on in-house synthesized reference compounds.

# Known murine urinary compounds.

*Tentatively identified.

**Unknown.

methods for further description. All non-significant fixed effects were dropped from the final models for parsimony.

The only significant effect on behavior PC1 was strain (Table 3). Tukey tests showed that SJL mice had the highest scores, followed by AJ, and then B6 mice (Table 3). Strain also significantly impacted behavior PC2 (Table 3): AJ mice had lower scores than B6 and SJL mice (Table 3). Urine PC3 had a positive effect on behavior PC2 ($F_{1,18.55} = 5.73$, $p_{adj} = 0.027$; $\eta^2 = 0.278$). Compounds with high loading on urine PC3 were β-farnesene, 5-ethylthiazolidine

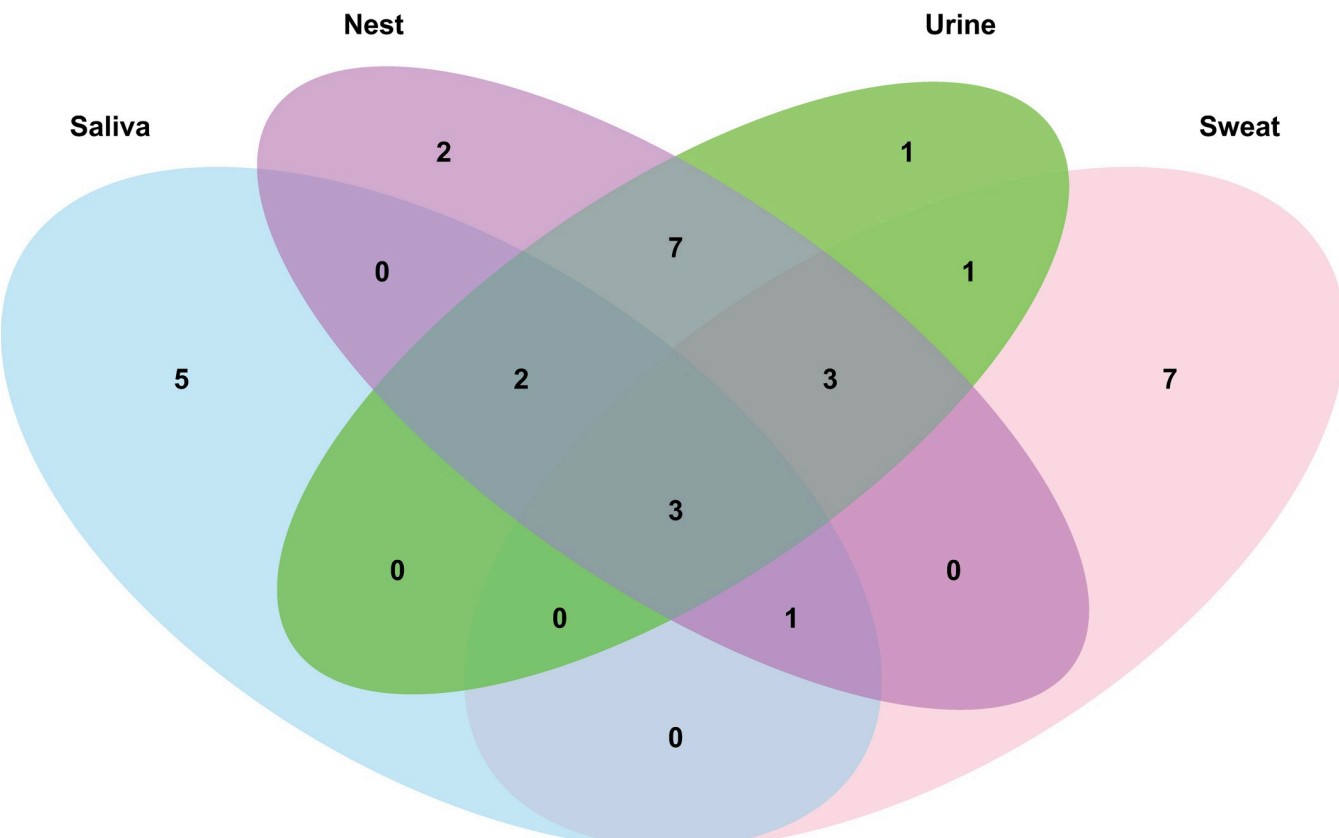

**Fig 1. Venn diagram of the number of volatile organic compounds detected in each sample type.**

(tentative identification), hexadecanol, and 2-isopropylthiazole related to the inter-male aggression promoting pheromone SBT [25] (Table 2).

Since the behavior PCs were primarily impacted by strain, historical social behavior patterns were confirmed in the featured strains to determine if they vary across study days. As expected, AJ, B6, and SJL mice displayed different levels of each social behavior: escalated aggression; mediated aggression; social investigation; allo-grooming; and group sleep (Table 4). Study day only impacted escalated aggression while the day*strain interaction was not significant for any behavior category.

After correcting for multiple comparisons, post-hoc custom tests showed that there was less escalated aggression on day 7 than 1 (GLIM: $\chi(1) = 5.88$, $p = 0.015$). SJL mice displayed more escalated aggression than AJ mice (GLIM: $\chi(1) = 7.95$, $p<0.005$), while post hoc Tukey tests showed SJL displayed the most mediated aggression (Tukey: $p<0.05$) and social investigation (Tukey: $p<0.05$). B6 and AJ mice displayed similar levels of all three behaviors (p values>0.05). B6 mice displayed the highest level of allo-grooming (Tukey: $p<0.05$) while SJL and AJ mice displayed similar levels (p>0.05). B6 also displayed more group sleep than AJ mice (Tukey: $p<0.05$), but SJL mice were similar to both strains (p values>0.05). All strain patterns are depicted in Fig 3.

Because strain had such an overwhelming effect on behavior, each of the tested VOC PCs and covariates were run in a mixed model to determine the impact of strain. VOC and genetic effects can both influence behavior, either independently or in conjunction with one another, which is why mixed models were used to examine whether strain influenced VOC PCs.

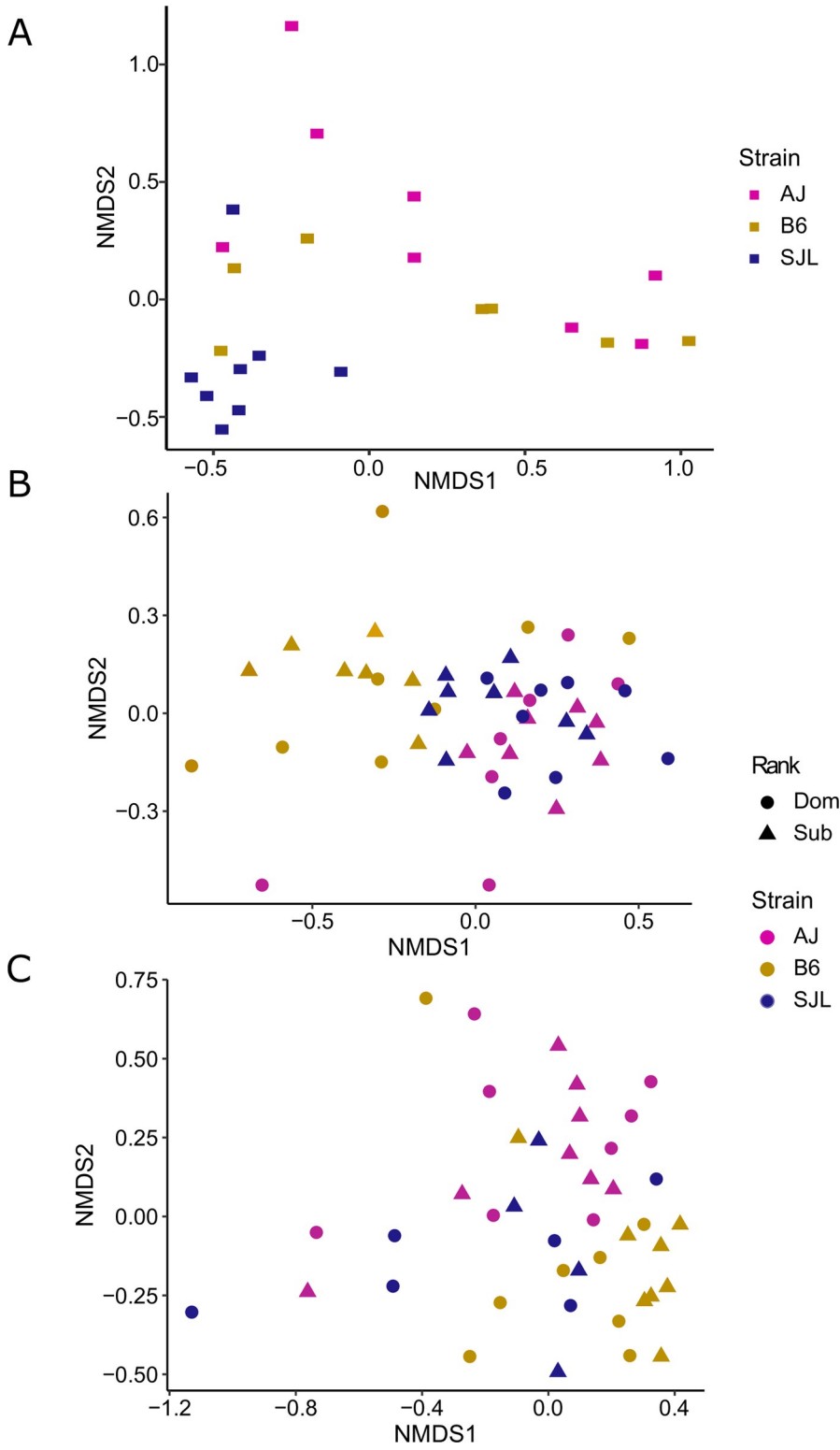

**Fig 2.** Volatile organic compound profiles of (A) nesting material, (B) plantar sweat, and (C) urine showed strain specific patterns. Non-metric multidimensional scaling using Bray-Curtis dissimilarity matrices for (A) used nesting material (stress = 0.095, N = 23), (B) plantar sweat (stress = 0.113, N = 46), and (C) urine (stress = 0.162, N = 42) showed sample separation corresponding to strain. Multivariate analyses using the Adonis test showed a significant difference in profiles between strains for all sample types: nest, p = 0.006; sweat, p = 0.001; and urine, p = 0.001. In

contrast, (B) sweat and (C) urine samples did not show separation based on social ranking and Adonis tests did not show significant profile differences.

Overall, strain had a significant effect on sweat PC1, nest PC1, urine PC1, and urine PC2 (Table 3). Post hoc Tukey tests showed SJL cages had lower nest PC1 scores and higher urine PC2 scores than B6 and AJ. B6 cages had higher scores on sweat PC1 and lower scores on urine PC2 than SJL and AJ. They also had higher scores on urine PC1 than AJ mice. AJ cages had lower scores on urine PC1 than B6, similar scores to B6 on nest PC1, and similar scores to SJL on sweat PC1. On urine PC2, AJ mice had higher scores than B6 and lower scores than SJL

**Table 2. Loading values for all Principal Components (PCs) retained for mixed models.**

| Sweat VOCs | Sweat PC1 | Nest VOCs | Nest PC1 | Nest PC2 | Urine VOCs | Urine PC1 | Urine PC2 | Urine PC3 | Behaviors | Behavior PC1 | Behavior PC2 |
|---|---|---|---|---|---|---|---|---|---|---|---|
| acetic acid | -0.2744 | acetic acid | 0.2644 | 0.2307 | acetic acid | **-0.3421** | -0.1522 | 0.1843 | Mediated aggression | **0.9248** | 0.2948 |
| hexadecanoic acid | -0.2521 | hexadecanoic acid | 0.2013 | **-0.3177** | hexadecanoic acid | 0.2033 | **0.3590** | -0.0798 | Escalated aggression | **0.9538** | 0.1987 |
| octadecanoic acid | -0.2592 | octadecanoic acid | 0.2174 | **-0.3018** | octadecanoic acid | 0.1254 | **0.4337** | -0.0655 | Allo-Groom | **-0.6467** | **0.6776** |
| 2-furanmethanol | -0.2429 | 2-furanmethanol | 0.2499 | -0.2432 | 2-furanmethanol | **-0.3654** | -0.1987 | 0.1066 | Social Invest. | **0.9341** | 0.2073 |
| 5-ethyl thiazolidine* | 0.1102 | 5-ethyl thiazolidine* | -0.0171 | -0.2847 | 5-ethyl thiazolidine* | -0.0252 | 0.1604 | **-0.5360** | Group Sleep | -0.2355 | **0.9246** |
| Hexadecanol | -0.1217 | hexadecanol | 0.1919 | -0.2664 | hexadecanol | 0.1329 | 0.0381 | 0.3545 | | | |
| geranylacetone | -0.2798 | geranylacetone | **0.3334** | 0.1998 | α-farnesene | 0.0874 | 0.1535 | 0.2034 | | | |
| 3-methyl-2(H)-furanone | 0.2838 | β-farnesene | -0.0437 | -0.0627 | β-farnesene | 0.0144 | 0.1638 | 0.3407 | | | |
| o-toluidine | 0.2145 | 1,2-cyclopentadione | **0.3004** | -0.0751 | 1 2-cyclopentadione | **-0.4024** | -0.1653 | 0.0867 | | | |
| 3,4-dimethyl-1,2-cyclo pentanedione | **0.3692** | dehydrobrevicomin | 0.0732 | **0.3475** | dehydrobrevicomin | 0.0034 | **0.3647** | 0.2565 | | | |
| N-formyl morpholine* | -0.1708 | 2-isopropylthiazole | 0.1520 | **0.3934** | 2-isopropylthiazole | -0.0710 | 0.2397 | **0.3703** | | | |
| indole | 0.2068 | 2-sec-butyl thiazoline | 0.1465 | **0.3691** | 2-sec-butyl thiazoline | -0.0745 | -0.1605 | 0.2979 | | | |
| ethylcyclo pentenolone | **0.3266** | 5,5-dimethyl-2-ethyl-4,5-dihydrofuran | -0.2729 | 0.1583 | 5 5-dimethyl-2-ethyl-4 5-dihydrofuran | **0.3689** | -0.2395 | 0.1111 | | | |
| 3,5-diethyl-2-hydroxycyclopent-2-en-1-one | **0.3362** | Z-5,5-dimethyl-2-ethylidene tetrahydrofuran | **-0.3078** | -0.0796 | Z-5 5-dimethyl-2-ethylidene tetrahydrofuran | **0.3652** | -0.1590 | 0.1483 | | | |
| methylcyclo pentenolone | 0.2737 | E-5,5-dimethyl-2-ethylidene tetrahydrofuran | **-0.3228** | -0.0493 | E-5 5-dimethyl-2-ethylidene tetrahydrofuran | **0.3407** | -0.1464 | 0.1601 | | | |
| | | 6-hydroxy-6-methyl-3-heptanone | -0.1421 | 0.2035 | 6-hydroxy-6-methyl-3-heptanone | **0.3013** | **-0.3145** | -0.0300 | | | |
| | | MW 152 compound** | **0.3653** | 0.0016 | o-toluidine | 0.1308 | -0.3097 | -0.1152 | | | |
| | | 2-hydroxy benzaldehyde | 0.2490 | 0.0823 | | | | | | | |

Loadings for sweat, nest, and urine VOCs were subjected to varimax rotation while loadings for behavior reflect original values. Values in bold were interpreted as high loading for each PC.

*tentative identification;

** unknown.

**Table 3. Strain patterns on VOC profile and behavior based on mixed models.**

| Dependent Variable | Strain Main Effect | Tukey Differences |
|---|---|---|
| **Behavior PC1** | $F_{2,18} = 256.62$, $\mathbf{p_{adj} < 0.001}$ | SJL > AJ > B6 |
| **Behavior PC2** | $F_{2,17.06} = 23.75$, $\mathbf{p_{adj} < 0.001}$ | (B6 = SJL) > AJ |
| **Nest PC1** | $F_{2,18.14} = 6.10$, $\mathbf{p_{adj} = 0.036}$ | (AJ = B6) > SJL |
| **Nest PC2** | $F_{2,17.52} = 0.85$, $p_{adj} = 0.886$ | --- |
| **Sweat PC1** | $F_{2,18} = 19.61$, $\mathbf{p_{adj} < 0.001}$ | B6 > (AJ = SJL) |
| **Urine PC1** | $F_{2,18} = 7.97$, $\mathbf{p_{adj} = 0.015}$ | B6 > AJ; SJL = B6; SJL = AJ |
| **Urine PC2** | $F_{2,18} = 20.05$, $\mathbf{p_{adj} < 0.001}$ | SJL > AJ > B6 |
| **Urine PC3** | $F_{2,18} = 0.02$, $p_{adj} = 0.983$ | --- |
| **Landau's H** | $F_{2,18} = 1.49$, $p_{adj} = 0.753$ | --- |
| **Nest Complexity Score** | $F_{2,18} = 148.74$, $\mathbf{p_{adj} < 0.001}$ | AJ > B6 > SJL |

$p_{adj}$ represents adjusted p values based on the Bonferroni sequential method. Significant p values for main effects are listed in **bold**. Specific differences between mouse strains was determined using post-hoc Tukey tests ($p < 0.05$). '---' indicates that a post-hoc test was not conducted due to the main effect not being significant.

(Table 3). Strain did not affect urine PC3 or nest PC2. In terms of covariate measures, strain significantly impacted average nest complexity score, but did not impact Landau's H. (Table 3). AJ mice built the most complex nests, followed by B6, and SJL (Tukey: $p < 0.05$).

In this study, the strain pattern of sweat PC1 matches that of allo-grooming, while patterns of nest PC1 and urine PC2 match that of aggression. Therefore, VOCs with high loading on these PCs were chosen for further analysis. Scores on sweat PC1 were positively correlated with both allo-grooming (Pearson's r = 0.66, 95% CI: 0.35–0.84, $p < 0.001$) and group sleep (Pearson's r = 0.52, 95% CI: 0.15–0.76, p = 0.011). The following compounds had high positive loading on sweat PC1: 3,4-dimethyl-1,2-cyclopentanedione, ethylcyclopentenolone, and a newly identified compound, 3,5-diethyl- 2-hydroxycyclopent-2-en-1-one (Table 2 and Figs 4C–4E and S1). A verified structure (Fig 4E) is related to ethylcyclopentenolone (Fig 4D). Of these, 3,4-dimethyl-1,2-cyclopentanedione and 3,5-diethyl- 2-hydroxycyclopent-2-en-1-one varied by strain and were correlated with allo-grooming; 3,5-diethyl- 2-hydroxycyclopent-2-en-1-one only was correlated with group sleep (Table 5).

Scores on nest PC1 were negatively correlated with both escalated (Pearson's r = -0.56, 95% CI: -0.79- -0.20, p = 0.005) and mediated aggression (Pearson's r = -0.49, 95% CI: -0.75- -0.10, p = 0.018). Compounds with high positive loading on nest PC1 were geranylacetone, 1,2-

**Table 4. Effects of strain and day on behaviors of interest based on mixed models.**

| | Strain | Strain comparison | Day | Day comparison |
|---|---|---|---|---|
| Escalated Aggression [a] | $\chi(2) = 8.06$, **p = 0.018** | SJL > AJ; B6 = SJL; B6 = AJ | $\chi(2) = 7.31$, **p = 0.026** | Day 1 > Day 7; Day 2 = Day 1; Day 2 = Day 7 |
| Mediated Aggression [b] | $F_{2,59.98} = 26.09$, **p < 0.001** | SJL > (AJ = B6) | $F_{2,42} = 0.73$, p = 0.486 | --- |
| Social Investigation [b] | $F_{2,50.86} = 19.71$, **p < 0.001** | SJL > (AJ = B6) | $F_{2,42} = 0.03$, p = 0.973 | --- |
| Allo-grooming [b] | $F_{2,52.43} = 43.91$, **p < 0.001** | B6 > (AJ = SJL) | $F_{2,42} = 0.59$, p = 0.557 | --- |
| Group Sleep [b] | $F_{2,57.85} = 5.56$, **p = 0.006** | B6 > AJ; SJL = B6; SJL = AJ | $F_{2,42} = 1.60$, p = 0.213 | --- |
| Nesting- paw [b] | $F_{2,55.26} = 3.21$, **p = 0.048** | AJ > B6; SJL = B6; SJL = AJ | $F_{2,42} = 2.01$, p = 0.147 | --- |
| Nesting- mouth [b] | $F_{2,39.31} = 4.48$, **p = 0.018** | AJ > SJL; B6 = AJ; B6 = SJL | $F_{2,42} = 0.41$, p = 0.663 | --- |

[a] analyzed with generalized linear mixed model and Bonferroni corrected contrasts ($p < 0.017$);

[b] analyzed with general linear mixed model and post-hoc Tukey test ($p < 0.05$); Significant p values are listed in **bold**. '---' indicates that a post-hoc test was not conducted due the insignificant main effect. The strain*day interaction was tested and not significant in any model.

cyclopentadione, and another unknown compound (Table 2 and Fig 5). We will refer to this unknown compound as MW 152 based on its assumed molecular weight. Currently the identity of MW 152 has not been determined. Two dehydration products of 6-hydroxy-6-methyl-3-heptanone had high negative loading on nest PC1 (Table 2 and Fig 5). Since positively loading compounds would be associated with less aggression, only they were analyzed. Geranylacetone was negatively correlated with both mediated and escalated aggression and varied by strain (Table 5). MW 152 was negatively correlated with escalated aggression and was not impacted by strain (Table 5).

Scores on urine PC2 were positively correlated with both escalated (Pearson's r = 0.63, 95% CI: 0.31–0.82, p<0.001) and mediated aggression (Pearson's r = 0.59, 95% CI: 0.24–0.80, p = 0.002). Compounds with high positive loading on urine PC2 were hexadecenoic acid, octadecanoic acid, and aggression-related dehydrobrevicomin [22], while testosterone dependent 6-hydroxy-6-methyl-3-heptanone [24] had high negative loading (Table 2). Negatively loading

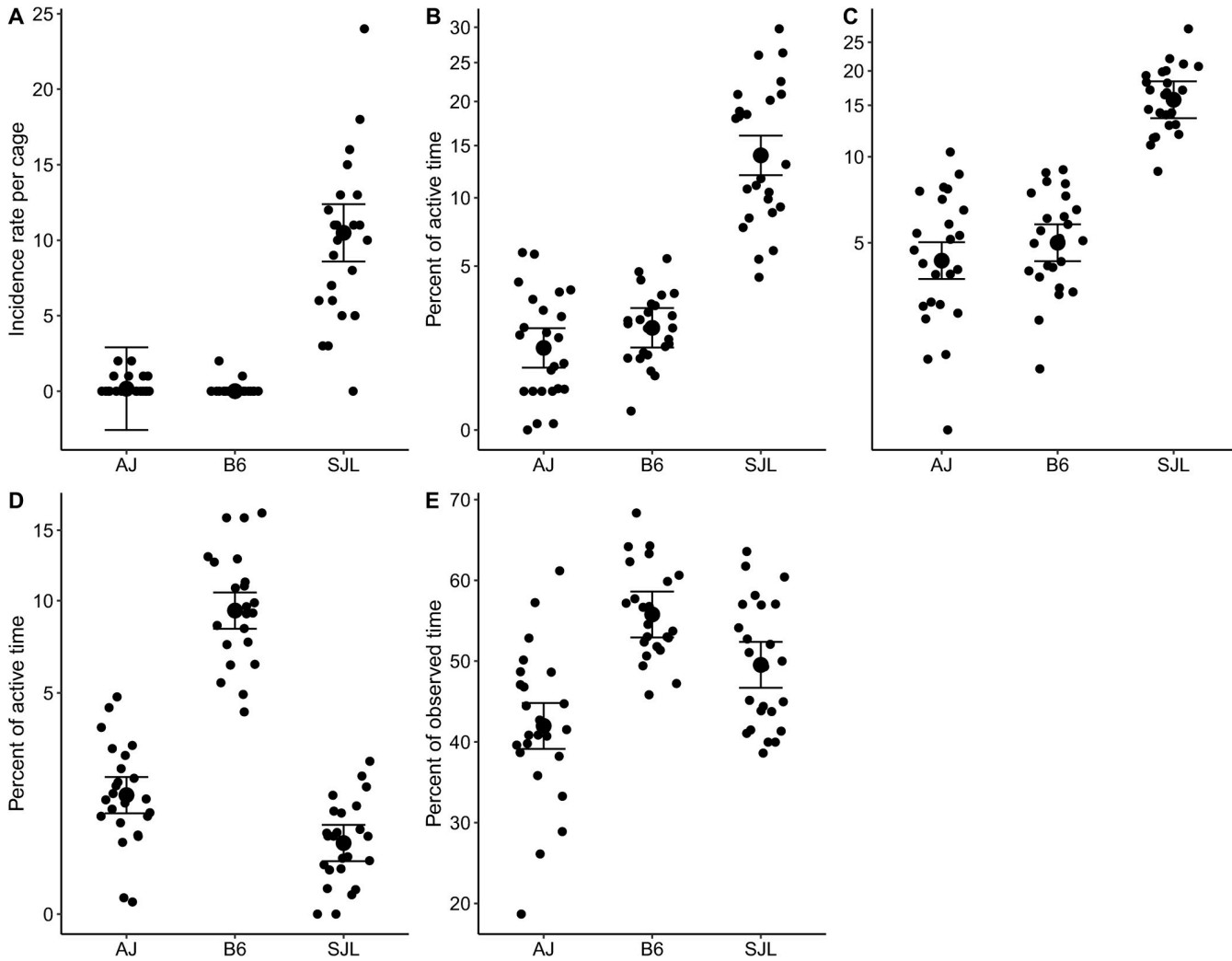

**Fig 3. Aggressive and affiliative behavior patterns varied according to strain.** SJL mice had (A) the highest rate of escalated aggressive behaviors (occurrences per day; p = 0.018). They also spent the highest percent of active time performing (B) mediated aggression (p<0.001) and (C) social investigation (p<0.001) behaviors. B6 mice spent the highest percent of active time (D) allo-grooming (p<0.001) and highest percent of observed time in (E) group sleep (p = 0.006). All data are presented as strain LSM +/- SE with the scatter of the individual data points' residual differences from the LSM (N = 72, 3 observations from 24 cages). Y axes are shown on a square root back transformed scale in B and D, and on a log$_{10}$ back transformed scale in C.

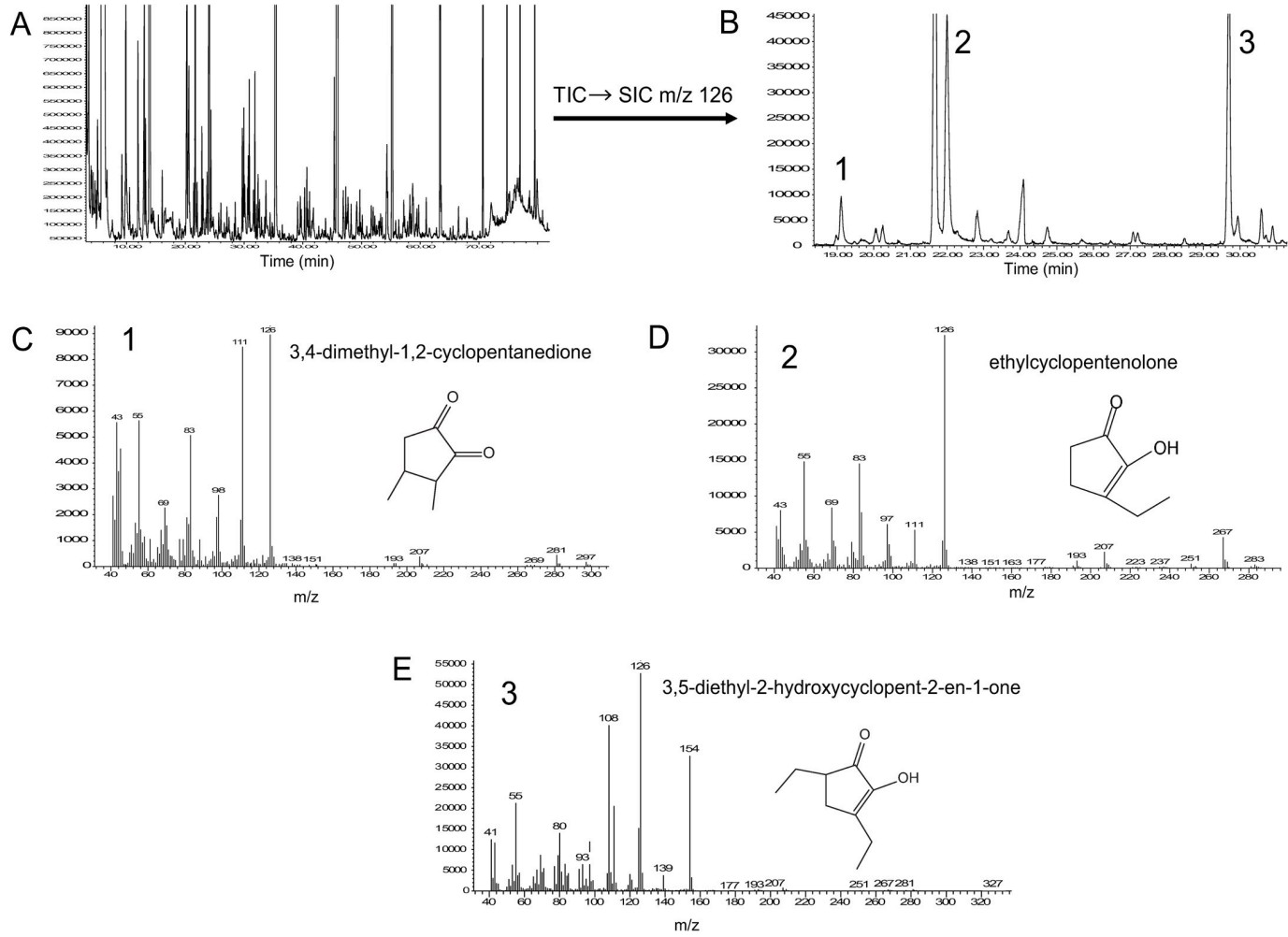

**Fig 4. High loading compounds on sweat PC1 and their mass spectra (EI 70 eV).** (A) Total ion chromatogram (TIC); (B) Post-run extracted m/z 126 single ion current chromatogram (SIC); (C) Compound 1, 3,4-dimethyl-1,2-cyclopentadione from SIC at retention time 16.1 min; (D) Compound 2, ethylcyclopentenolone from SIC at retention time 22.02 min; (E) 3,5-diethyl- 2-hydroxycyclopent-2-en-1-one from SIC at retention time 29.69 min.

compounds would be associated with less aggression, so only 6-hydroxy-6-methyl-3-hepta-none was further analyzed. It was correlated with allo-grooming and varied by strain (Table 5).

Since urine PC3 significantly impacted behavior PC2, it was also compared to each behavior. Urine PC3 did not correlate with any individual behaviors or show strong strain variation (Table 3), so high loading compounds were not examined further.

In summary, SJL mice displayed substantially more aggressive behavior and social investigation. They also had the highest scores on urine PC2 and the lowest on nest PC1. B6 mice displayed the most allo-grooming, had the highest scores on sweat PC1, and the lowest on urine PC2. AJ mice displayed minimal social behavior, performed the most nesting behavior, and had the highest nest complexity scores (Fig 6).

## Strain and nesting behavior

A side objective of this study was to explore how nest manipulation behaviors varied across the three strains, since the main focus examined secreted chemical contents on nesting material resulting from manipulation with the paws or mouth. Separate mixed models were run for

**Table 5. Relationship between behavior, strain, and high loading VOCs from sweat PC1, nest PC1, and urine PC2.**

| VOC | Odor PC | Behavior correlation | Strain | Strain comparison |
|---|---|---|---|---|
| **3,4-dimethyl-1,2-cyclopentanedione** | Sweat PC1 | **Allo-grooming**: Pearson's r = 0.58, 95% CI: 0.23–0.80, **p = 0.003** | $F_{2,18}$ = 14.66 **P<0.001** | B6 > (SJL = AJ) |
| **ethylcyclopentenolone** | Sweat PC1 | NS | $F_{2,18}$ = 1.07 P = 0.364 | --- |
| **3,5-diethyl- 2-hydroxycyclopent-2-en-1-one** | Sweat PC1 | **Allo-grooming**: Pearson's r = 0.62, 95% CI: 0.29–0.82, **p = 0.001** Group sleep: Pearson's r = 0.54, 95% CI: 0.17–0.77, **p = 0.007** | $F_{2,18}$ = 8.27 **P = 0.003** | B6 > (SJL = AJ) |
| **geranylacetone** | Nest PC1 | **Escalated aggression**: Pearson's r = -0.52, 95% CI: -0.77- -0.13, **p = 0.011** **Mediated aggression**: Pearson's r = -0.43, 95% CI: -0.72- -0.02, **p = 0.04** | $F_{2,17}$ = 4.85 **P = 0.022** | SJL < AJ; B6 = AJ; B6 = SJL |
| **1,2- cyclopentadione** | Nest PC1 | NS | $F_{2,17}$ = 0.87 P = 0.435 | --- |
| **MW 152** | Nest PC1 | **Escalated aggression**: Pearson's r = -0.41, 95% CI: -0.71- -0.001, **p = 0.05** | $F_{2,17}$ = 2.76 P = 0.091 | --- |
| **6-hydroxy-6-methyl-3-heptanone** | Urine PC2 | **Allo-grooming**: Pearson's r = 0.60, 95% CI: 0.25–0.81, **p = 0.002** | $F_{2,18}$ = 19.48 **P<0.001** | B6 > (SJL = AJ) |

Significant p values are listed in **bold**. 'NS' indicates no significant correlations detected. '---' indicates that a post-hoc test was not conducted due the insignificant main effect.

manipulation performed with the paws and mouth. Behaviors performed with the paws were expected to influence compounds originating in the sweat while behaviors performed with the mouth would have more impact on compounds from the saliva. Nesting done with the paws and mouth were significantly influenced by strain (Table 4). Post hoc Tukey tests showed that AJ mice performed more nesting with their paws than B6 (Tukey: p<0.05), while SJL were similar to both (p values>0.05). AJ mice performed more nesting with their mouth than SJL (Tukey: p<0.05), while B6 were similar to both (p values>0.05).

## Discussion

To our knowledge, this experiment is the first to report the VOC profiles of used nesting material and foot plantar gland sweat in male laboratory mice (aim 1). It is also the first to examine the relationship between these profiles and social behavior (aim 2). It has been shown that preserving used nesting material can reduce aggression at cage change [12], but the theory that nesting material holds aggression reducing plantar sweat has remained speculation until now.

### Observed behavior

The behavior PCA, PC1 showed that mediated aggression, escalated aggression, and social investigation were strongly correlated across all cages. In contrast, allo-grooming was negatively associated with the latter three behaviors on PC1, and positively associated with group sleep behavior on PC2. However, all of these patterns were strongly explained by strain. On behavior PC1 (high aggression and low allo-grooming), SJL had the highest scores followed by AJ and then B6. This reflects the greater amount of aggression and social investigation performed by SJL and the greater amount of allo-grooming performed by B6. On the other hand, behavior PC2 scores (high allo-grooming and group sleep) reflect the higher amount of group sleep performed by B6 and SJL mice than AJs.

Several of these strain patterns were unexpected. First, SJL mice are known for excessive inter-male aggression [38], but they also displayed the most social investigation behavior. Our coding scheme was not detailed enough to make conclusions about the direct behavioral sequence, but anecdotally, social investigatory sniffing tended to precede aggressive

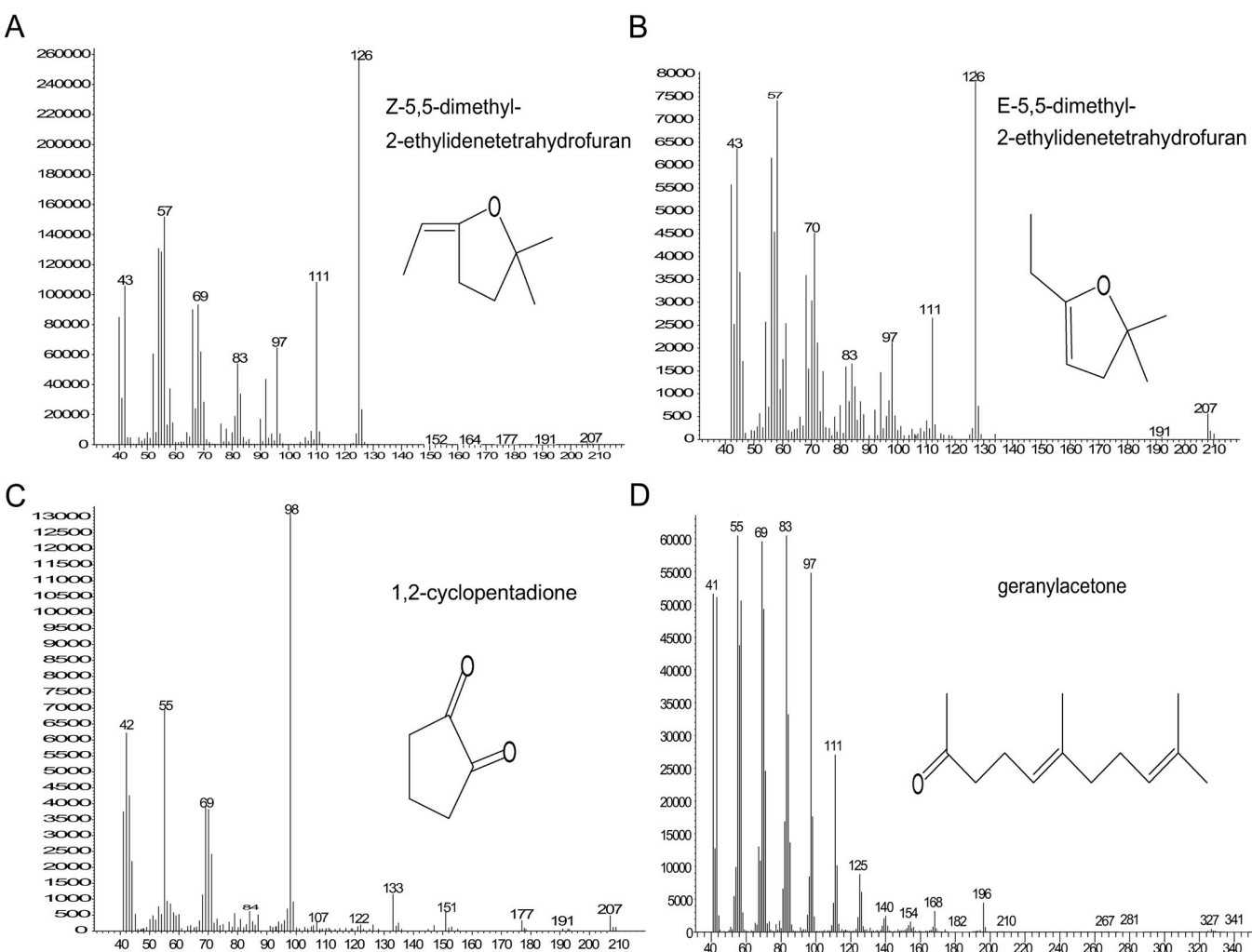

**Fig 5. High loading compounds on nest PC1 and their mass spectra (EI 70 eV).** (A) Z-5,5-dimethyl-2-ethylidenetetrahydrofuran at retention time 7.98 min; (B) E-5,5-dimethyl-2- ethylidenetetrahydrofuran at retention time 9.38 min; (C) 1,2-cyclopentadione at retention time 10.85 min; (D) geranylacetone at retention time 43.93 min.

interactions. Initially the ethogram did not include a separate category for social investigation, but it was added after observing this pattern after the first few cages. This calls into question the underlying motivations of sniffing behavior, as it is traditionally considered to be neutral or exploratory [39–41]. However, these data make the actor mouse's intentions less clear.

Second, B6 males are frequently the subject of caretaker complaints about aggression. Here they displayed minimal aggression which is consistent with previous work [42, 43], but we anticipated conflict in some cages in order to demonstrate a more linear relationship between VOCs and observed aggression. Thirdly, AJ cages displayed minimal social interactions, aside from group sleep. They are known for minimal levels of inter-male aggression [38], so we mistakenly presumed that this would equate to higher rates of affiliative behavior. Generally, aggressive and affiliative behaviors are performed more by species that are sociable, like mice [5]. However, AJ have previously demonstrated low sociability to stranger mice [44, 45], so these data extend this pattern to behavior towards familiar cage mates.

We purposefully designed this experiment to incorporate multiple inbred mouse strains in order to ensure that a wide range of specific behaviors were observed. However, we did not

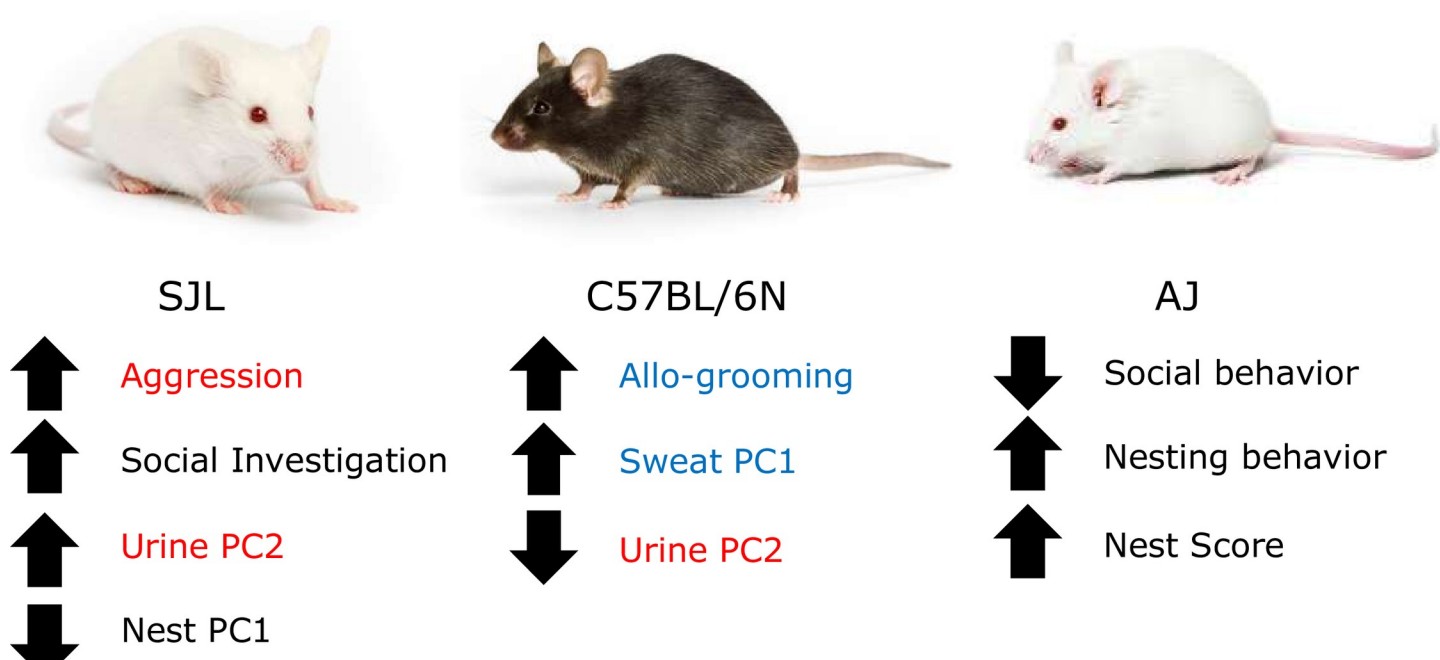

**Fig 6. Summary diagram of observed strain patterns.** Variables that were associated with aggressive behavior are listed in red, while those associated with an affiliative behavior are listed in blue.

expect to find such limited variation within these strains. Thus, strain unfortunately acts as a confounding factor for subsequent interpretations.

## VOC patterns that match behavior

Overall, we found that several VOCs in urine, sweat, and nesting material aligned with strain specific patterns of social behavior in the home cage. While VOCs did not directly account for a significant amount of variation in aggressive behavior, it is possible that they may be one of the many factors that contribute to inherent strain differences in behavior. Along with relatively high aggression levels, SJL mice displayed low scores on nest PC1, and high scores on urine PC2. Scores on each of these respective PCs were negatively and positively correlated with aggression. Therefore, VOCs with a positive loading on nest PC1 and a negative loading on urine PC2 showed potential for an aggression appeasement role.

Geranylacetone was the only VOC from the nest to be both negatively correlated with aggression and have a strain specific pattern. It was produced less in SJL mice than AJ, but quantities in B6 were similar to both other strains. It was also present in sweat and saliva samples and has previously been detected in hamster ventral glands [46]. This gland is typically used for territory marking [47] and there is some evidence that secretions are capable of changing in response to individual social interactions [46]. Perhaps proportions from the nest samples related to aggression due to a dilution effect from being in the environment. Odor signals are often effective at small concentrations, so the values seen here from pure body fluids may be too high to relate to behavior. Additionally, quantities of geranylacetone showed the same strain pattern as nest complexity score. Scores were lowest in SJL, which supports previous research showing that nest score decreases with the number of wounded mice in a cage [48]. As mice engage in more aggressive interactions that include rapid fighting or chasing,

any existing nest structure is likely to be destroyed during escape attempts. As stress and pain levels rise in the cage, motivation to restructure or maintain a complex nest decreases [48].

MW 152 was negatively correlated with escalated aggression and was the only compound not detected in any of this study's body fluid samples. Furthermore, it was not present in control (unused) nest samples, so it is possible that it originated from another body gland, fur oils, or fecal residues. Although precautions were taken to minimize contamination with fecal residue (e.g. cleaning the surface of the foot and the anesthesia chamber), it is possible that fecal odors could have contaminated the samples and future work could examine how the fecal VOC profile may impact behavior. At this time, a verified structure for MW 152 has not been determined.

The only VOC with a high negative loading on urine PC2 was 6-hydroxy-6-methyl-3-heptanone, a MUP ligand that accelerates puberty in female mice [49]. Although it did not directly relate to aggression, it was positively correlated with allo-grooming and was produced more by B6 mice than SJL and AJ. This result was unexpected since male mouse pheromones from urine have been shown to promote aggression between males [25, 50]. However, to our knowledge, 6-hydroxy-6-methyl-3-heptanone has not been directly tested for effects in males. Based on this data, it may have a role promoting affiliative behavior.

Although this study aimed to find aggression reducing compounds, the relationship between sweat and social behavior was central to the study hypothesis. B6 mice had the highest scores on sweat PC1 and displayed the most allo-grooming. Of the VOCs with a high positive loading, 3,4-dimethyl-1,2-cyclopentanedione and 3,5-diethyl- 2-hydroxycyclopent-2-en-1-one both were correlated with allo-grooming and were produced in higher quantities by B6 mice than AJ and SJL. 3,5-diethyl- 2-hydroxycyclopent-2-en-1-one was also correlated with group sleep. To the best of our knowledge, 3,4-dimethyl-1,2-cyclopentanedione does not have a known behavioral role, but shows potential for improving mouse welfare. It would be a worthy candidate for future behavioral testing to explore its potential role in mouse communication, along with the newly discovered 3,5-diethyl- 2-hydroxycyclopent-2-en-1-one. To our knowledge, these kinds of cyclopentanone derivatives are unique to mouse plantar sweat and, based on our data, may play a role in promoting affiliative behaviors.

## Dominance hierarchy

Surprisingly, dominance linearity in the tube test as measured by Landau's H did not account for significant differences in behavior. This result is the opposite of what we had expected. A previous study showed that increasing values of Landau's H correlated with lower levels of aggression, suggesting that certainty in social rank reduces escalated interactions [51]. One main difference between that study and this was that the former used outbred CD-1 mice, while inbred strains were used here. This may reflect a strain impact on the relationship between dominance linearity in the tube test and aggression. Additionally, the previous study measured aggression and linearity during multiple time periods and across changes in cage enrichment. Our study focused on a one-week time period and kept housing conditions stable. Even though mice were acclimated to the arena before testing, it has been argued that there is a learned component to tube testing, such that more than one tube testing session is required for mice to display valid rankings outside the home cage [52]. However, a previous assessment of stable male groups found the tube test produced inconsistent rankings over 3 weeks' time, with the most stable relationships occurring between the second and third trials [53]. This finding was published while our experiment was in progress; consequently, the approach used here does not take these new findings into consideration. It was also suggested that competitive learning in the tube test may be specific to that arena and not reflect home cage behavior [53].

Considering both the contrasting relationship between dominance linearity and aggression, and the lack of variation in β-farnesene and SBT between dominant and subordinate urine, it is likely that the tube test, at least as it was carried out here, may not be a valid indicator of individual in-cage social rank. That being said, the lack of a relationship between rank stability in the tube arena and aggression in the cage may still be meaningful. Further research will be valuable in explaining differences between tube test social rank and in cage social rank.

## Limitations and future research

In this study, we were concerned about obtaining a sufficient quantity of sweat for analysis and utilized pilocarpine injections to increase sample volume. We do acknowledge that using pilocarpine to induce plantar sweat secretion may have unknown effects on VOC ratios. Pilocarpine functions by stimulating M3 muscarinic receptors on exocrine glands, such as the sweat glands [54]. Currently, there is little evidence to determine how the increased gland activity impacts VOC content, but it is possible the compounds were diluted in the larger sample volume. Work in humans shows that sweat induced by pilocarpine is generally similar in content to sweat induced by exercise, although the latter contains more compounds indicative of a more demanding metabolic state [55]. However, mice do not produce sweat to thermoregulate, and to the best of our knowledge, there are no direct VOC comparisons of fluids collected without stimulation versus pilocarpine. Additionally, individual variation in responses to the pilocarpine treatment could have impacted the data. Pilocarpine is a common treatment for dry mouth in humans, but efficacy can depend on the individual [56]. At this time, factors that impact pilocarpine success have not been identified, and it was not possible to quantify the volume of collected sweat based on the sampling method.

A second limitation worth noting is that this study only focused on VOC profiles. It is possible that protein signals could have impacted these data. Urinary MUP20 ("darcin") in particular is a pheromone that promotes aggression between males, but also is necessary for social learning to occur [26, 57]. Darcin is expressed more in mice of the C57 lineage [58], so it is possible that it caused B6 mice to become familiar with cage mates more quickly than other strains and as a result perform more affiliative behavior. That being said, production itself cannot predict aggression since AJ and SJL mice both produce low levels of darcin [59], but more complex compound interactions have yet to be explored. This study also did not address the effect on behavior of individual differences in odor perception. Many odor signals, especially pheromones, are detected by the vomeronasal organ (VNO) [60]. Gene expression in the VNO, particularly those encoding chemoreceptors, show great variation between strains and could be a major contributor to variability in behavior [61]. While strain specific expression was the focus of this study, we cannot assume that sensitivity follows the same pattern. For example, even though darcin is produced more by the C57 line, BALB/c males (Castle lineage) are still reactive and display the expected scent marking response when exposed to it [62].

Another point of consideration in this study was the amount of time mice in each cage spent performing nesting related behaviors, as this is likely to impact the relative amount of VOC deposits in the nest. AJ mice performed the most nesting done with the paws and mouth, but their scores on nest PC1 were similar to those of B6. Of the high loading compounds on nest PC1, geranylactone was detected in both sweat and saliva samples, and 1,2-cyclopentadione (tentative) was detected in saliva. However, all of the VOCs in the nest samples traced to sweat or saliva were also detected in urine. Since the VOCs in the nest deposits are produced in multiple body fluids, it is difficult to conclude how time spent nesting directly impacted the nest VOC profile. This is especially true since our saliva samples were not sufficient for quantitative analyses. Anecdotally, AJ mice produced the lowest volume of

saliva, so the increased time spent nesting may be necessary for compounds levels on the nest to be similar to B6.

## Conclusion

Overall, this study found that, in the home cage, odor profiles from sweat, nesting material, and urine, show strain specific patterns that align with affiliative and aggressive behavior. These findings warrant future studies that directly test the influence of compounds found in sweat, urine, and nesting material on expression of social behaviors, to hopefully put the field one step closer to promoting socio-positive behaviors and improving laboratory mouse welfare.

## Methods

### Ethics statement

All procedures were approved by Purdue University's Institutional Animal Care and Use Committee (protocol #1707001598) and reporting adhered to the ARRIVE 2.0 guidelines [63]. The protocol was not previously registered before conducting the study.

   Due to concern over heightened aggression in the cage, we established humane endpoint criteria in which any mouse with wounding greater than 1cm$^2$ would be immediately euthanized. Animals were monitored daily for general activity and signs of pain/distress. If any animals developed minor wounding, they were monitored more frequently. No mice reached our endpoint criteria.

### Animals

All mice in this study were acquired from Charles River and were free of common known pathogen agents at shipping. More information can be found in [64]. Eight cages each containing five male mice of the following strains were used: SJL/JOrlIcoCrl (SJL)- Wilmington, MA; C57BL/6NCrl (B6)- Kingston, NY; and A/JCr (AJ)–Frederick, MD (N = 24 cages; 120 mice). Per the ARRIVE 2.0 guidelines [63], we are declaring that no strain served as a traditional control due to the study's exploratory nature. Sample size was determined using Mead's resource equation. Due to spatial constraints, the twenty four cages were divided into four equal groups containing two cages per strain. B6 mice were used as they are the most commonly studied inbred mouse and have the widest practical application; SJL males were used as a known high-aggressive strain [38]; while AJ mice were used as a known low aggressive strain compared to B6 mice [65]. Mice arrived at approximately 8 weeks of age and were housed for one week in open top micro-isolator cages, 11.5" x 7.25" x 4.25" (Ancare, Bellmore, NY) with food (Envigo, Teklad 2018, Indianapolis, IN) and reverse osmosis water offered *ad libitum*. Each cage contained aspen wood chip bedding (NEPCO, Warrensburg, NY) and 8.5g of virgin kraft crinkle paper (Enviro-dri, Fibercore, Cleveland, Ohio) for nesting material. Cages were kept under a 12:12 light: dark cycle (lights on at 06:00) with relative humidity ranging between 28–76% and temperature ranging between 18.8–23.3˚C. All mice were weighed at the beginning (mean weight 20.06 ±1.71 g) and end (mean weight: 21.73±1.86g) of the study and ear punched for identification. All animal handling was performed by female researchers and husbandry staff. Male scents can influence stress response in rodents and may alter baseline measurements [66].

   Upon arrival, mice were randomly distributed into the cages (5 mice per cage) from the shipping containers using a numerical sequence from RANDOM.org. Cage placement on the two MetroRacks was initially randomized based on a RANDOM.org sequence, and

subsequently balanced by strain across two shelves on each rack. Each shelf contained 2 cages and was enclosed by partitions of white foam board (Office Depot, Boca Raton, FL) to remove background noise for video monitoring (see Home Cage Observation below). Light intensity during the day was reduced from 430 lux, in the middle of the room, to an average of 67 lux at each cage location. Each cage was given its own numerical label from 1 to 24 that corresponded to its group and strain. Only the numerical label was present on the cage card to partially blind caregivers to cage identities during routine husbandry and research staff during sample collection/processing, behavior tests, and video coding.

## VOC sample collection and processing

**Nest.**    Mice were left in their home cage for 7 days after arrival. At the end of the week, 25 strips of crinkle paper were collected for VOC analysis (see below for GC-MS procedure). Samples were taken from both the periphery and center of the nest since mice restructure their nests daily [67] and it is not known if they are in contact with one area more than another. Some cages did not contain a structured nest, so the area containing dispersed material was divided into quadrants and each quadrant was equally sampled. The weighed sample of crinkle paper was placed in a 10 mL head-space sample vial with a Teflon cap (Gerstel GmbH, Mülheim an der Ruhr, Germany). An acetone (Avantor, Center Valley, PA) washed, straightened, and dried metal paper clip was punched through the vial Teflon seal. A magnetic Gerstel stir bar was attached to the clip above the nest material, 5 μL of 7-tridecanone in methanol (Baker Analyzed, Mallinckrodt Baker Inc., Phillipsburg, NJ) (8 ng/5 μL) was added to the nest material and the vial cap was closed tight. The head-space VOCs were collected at room temperature for 1 hour.

Two exceptions occurred within the AJ strain during nest sample collection. One cage flooded at the end of the third study day. Nest material was soaked and unable to be collected. It was replaced and subsequently collected four days later. A second cage flooded on the sixth study day. The nest from this cage was collected since there was a short proximity to the planned sampling day and enough dry material could be collected for processing. The former data point produced unusual data and was excluded from analysis; however, the latter was included.

**Sweat.**    To analyze compounds from mouse sweat, the stir bar surface sampling method (previously used for human skin VOC analyses) was replicated [68, 69]. To collect secretions from the plantar sweat glands, mice were anesthetized with compressed isoflurane and each foot was cleaned with ethanol. After air drying, hindfeet and forefeet were given a subcutaneous injection of 50 μL and 20 μL of 1mg/1mL pilocarpine (Sigma- Aldrich, St. Louis, MO) respectively. Previous studies have shown that gland activity is highest approximately 10–20 minutes after injection [70, 71], so mice were kept under anesthesia for 20 minutes post injection. Sweat was collected on the surface of one forefoot and one hindfoot per mouse using Twister™ polydimethylsiloxane coated stir bars (Gerstel GmbH, Mülheim an der Ruhr, Germany) embedded previously with the internal standard, 7-tridecanone (Sigma- Aldrich, St. Louis, MO) as described previously [69]. Every five minutes post injection the stir bar was rolled across the surface of the hind and forefeet five times. All collections were performed in the mice's housing room between the 7th and 9th hour of the light cycle. All mice were monitored throughout the procedure for signs of distress (uneven, shallow breaths; pale color of foot tissue).

**Saliva.**    Saliva was collected while the mice were anesthetized for sweat collection as the pilocarpine injections also stimulated saliva production. After the mice lost consciousness, the exposed chamber floor was quickly cleaned with ethanol. Saliva samples were collected via

pipette from the acrylic chamber floor and transferred into a 1.5 mL centrifuge tube. Saliva samples (25–100 μL) were pipetted into 20 mL glass scintillation vials containing 5.0 mL water (OmniSolv™ LC-MS grade, EMD Millipore Corporation, Billerica, MA), 8 ng of 7-tridecanone as an internal standard and the Twister™ stir bar. The vial was placed in a water bath at 40˚C for 2.5 hours for static aqueous stir bar extraction. This sampling method was modified from a previously reported study with human saliva [72].

**Urine.**   Since mice naturally urinate upon handling, each mouse was held over a fresh aluminum foil bowl to collect urine on day 5 of the study week, before behavior testing. Gentle abdominal massage was administered when needed to facilitate collection and samples were transferred via pipette to a 1.5mL centrifuge tube. However, when mice would not urinate during handling, the fluid was collected after the mice acclimated to the plexiglass tube test arena used for the behavioral assay (see Social Ranking section for description).

Urine samples (15–200 μL) were pipetted in a 20 mL glass scintillation vial with the metal foil cap containing 2.0 mL of water (OmniSolv™) [73], 8 ng of 7-tridecanone as an internal standard and a Twister™ stir bar. Stir bar extraction was performed for 60 min at room temperature at 850 rpm speed (15-place stir plate Variomag Multipoint HP15, H+P Labortechnic, Oberschleissheim, Germany).

After extraction, all stir bars were washed with OmniSolv™ water, dried with non-lint Kim-Wipes tissue (Kimberly-Clark, Roswell, GA), and placed in a Thermal Desorption Autosampler and Cooled Injection System (TDSA-CIS 4 from Gerstel GmbH) connected to an Agilent 6890N gas chromatograph– 5973iMSD mass spectrometer (Agilent Technologies, Inc., Wilmington, DE).

Since the sampling unit was the cage, sweat and saliva samples were collected from each cage's dominant and subordinate mouse based on results from the tube test (see Social Ranking section for test procedure) as social ranking has been reported to impact pheromone levels [20, 49, 74, 75]. Urine was collected from each mouse, but only samples belonging to each cage's dominant and subordinate were analyzed. All samples were collected at Purdue University and transported to Indiana University for analysis. In total, 24 nest samples, 48 sweat samples, 48 saliva samples, and 42 urine samples were collected. Six mice, each from a different cage of the SJL strain, did not produce urine when stimulated. Additionally, two sweat samples originating from different cages lost their labels during transport and could not be processed, leaving 46 data points for sweat analysis.

## Gas Chromatography-Mass Spectrometry (GC-MS) analysis

Splitless mode was used for thermal desorption sampling (TDS) with a temperature program of 20˚C for 0.5 min, then a 60˚C/min increase up to 280˚C for 8 min. The transfer line temperature was set at 290˚C and the cooled injection system (CIS) was cooled using liquid nitrogen to 0˚C during the thermal desorption. For the sample introduction into the GC-MS, the CIS was heated at 12˚C/s to 280˚C and held for 10 min. Solvent vent mode was used for the CIS inlet with a vent pressure of 9.1 psi, a vent flow of 50 mL/min, and a purge flow of 50 mL/min. The gas chromatograph (GC) separation capillary was a DB-5MS (30 m x 0.25 mm, i.d., 0.25 μm film thickness) from Agilent, and the GC carrier gas (helium) head pressure was 9.1 psi at a constant 1.2 mL/min flow mode. The GC oven temperature program started at 40˚C for 1 min, then increased at 2˚C/ min to 180˚C and immediately 10˚C/ min to 230˚C and held for 6 min (total GC run time 85 min). For the mass spectrometer (MS), positive electron ionization (EI) mode at 70eV was used with a scanning rate of 2.47 scans/s and mass range of 41–350 amu. The mass spectrometric detector (MSD) transfer line temperature was 300˚C, the ion source temperature was 230˚C, and the quadrupole temperature was set at 150˚C.

Compounds were identified or tentatively identified by matching retention times and mass spectra with standard compounds when available (Sigma-Aldrich Chemical Co.) and with spectra through NIST Mass Spectral Search Program for the NIST/EPA/NIH Mass Spectral Library (Version 2.0 a, 2002). Additionally, in-house (Novotny Laboratory) synthesized mouse urinary pheromone compounds and the in-house spectral database were utilized for identifications.

All VOC data was used to calculate odor proportions by dividing each absolute peak value by the sample's total peak area [76]. This was done to determine how behavior is affected by the relative VOC amount perceived by the mice. Due to the low volume of saliva that was collected, the GCMS analysis was unable to provide reliable quantitative values. The saliva VOC profile only served to make qualitative comparisons about nest compound origins. All VOC data was used to calculate odor proportions by dividing each absolute peak value by the sample's total peak area [76]. This was done to determine how behavior is affected by the relative VOC amount perceived by the mice. Due to the low volume of saliva that was collected, the GCMS analysis was unable to provide reliable quantitative values. The saliva VOC profile only served to make qualitative comparisons about nest compound origins.

## Behavioral measures

**Home cage observations.** Cages were continuously recorded for one week from arrival to sample collection with closed circuit television cameras (Sony, Tokyo, Japan) and GeoVision monitoring software (Taipei, Taiwan). Dark cycle recordings used 2 infrared illuminators (Sodial, China) per cage. The following social behaviors were documented: escalated aggression, mediated aggression, allo-groom, group sleep, and social investigation (Table 6). Coders were partially blinded to strain due to the difference in coat color between B6 and AJ/SJL. All social interactions were scored using one-zero focal sampling for one minute every five minutes between 12:00AM- 12:00PM on days 1, 2, and 7 of the study.

Since we were interested in compounds deposited on the nest, we were also interested in how mice interacted with the nest. Thus, oral nest manipulation and paw nest manipulation (Table 6) were scored using one-zero sampling for one minute every half hour between 12:00AM- 12:00PM on days 1, 2, and 7 of the study.

The 12:00AM -12:00PM time frame was chosen because it allows for equal observation across light and dark conditions and the mice experienced the least amount of disturbance during this time frame. Day 1 was monitored to include behaviors that occurred while the mice adjusted to their new cage, before the hierarchy is established; day 2 reflects interactions that occur as the hierarchy is beginning to form; and day 7 reflects the last 24 hours of the study in which the hierarchy is established [77]. Day 7 is also a common day for mice to undergo cage cleaning, so the maximum level of secretions in the nesting material represents the amount that many mice are exposed to before their nests are replaced. Ultimately the proportion of active time in which each behavior category occurred was determined for each cage, with the exception of group sleep for which the proportion of all observed time was calculated.

**Nest scores.** Daily nest scores were taken around the ninth hour of the light cycle based on Hess et al. [78]. This time was used as it is when nest scores are typically highest [67]. This scale was used as it provides the most variability for mice that are good nest builders and has been shown to reflect changes based on aggression [48]. Briefly, the nest is divided into a square region and each quarter is given a score from 1–5 based on its complexity with higher scores corresponding to more complex structures. The four quarter scores are then averaged for the overall nest score of a cage. In situations where more than 1 nest was present in a single

**Table 6. Ethogram of observed behavior categories.** All descriptions were taken from mousebehavior.org.

**Social Behaviors- recorded every 5 minutes using one-zero sampling**

| Category | Behavior | Description |
|---|---|---|
| Mediated Aggression | Resource Theft | A mouse will approach another that is either eating a piece of food or chewing on a piece of bedding. The approaching mouse will then attempt to take the resource from the other's paws or mouth. It may or may not be successful. It is often preceded by facial sniffing and involves one or both mice tugging at the resource. |
| | Tail Rattling | Fast waving movements of the tail. This behavior may be partially obscured by bedding material, but can be detected by displacement of bedding near a mouse's tail. |
| | Thrust | The aggressor mouse will first threaten its target cage mate by thrusting its head and fore body towards its cage mate's head or body. The aggressor's paw may come in brief contact with the target, but otherwise no contact is made. |
| | Mounting | Attempts to mount another animal in the absence of intromission. Palpitations with forepaws and pelvic thrusts may be present. |
| | Chase | A mouse will chase a fleeing partner, but no biting occurs |
| | Submissive Upright | A posture where the animal will sit on its haunches in an upright position exposing the belly. The forepaws are off the ground and the mouse may stretch out its forepaws towards the threatening mouse. Mouse can also be laying on its side with one forepaw and one hind paw stretched toward the threatening mouse and its back touching the ground. |
| | Fleeing | This behavior is characterized by a mouse moving away from the mouse performing an aggressive behavior. Typically fleeing animals will run, but in a confined space may walk or turn first. Also score if the mouse turns away without locomoting. Only score if responding to an aggressive behavior (mediated/escalated) or investigation. |
| Escalated Aggression | Bite | The aggressor mouse attacks the recipient with open mouth and appears to bite the recipient, or latches onto the recipient by his teeth, or forcefully touches the recipient who responds by jumping or fleeing quickly. This also includes a mouse using its teeth to grab and tug on another's tail. Only score for the mouse that is biting. |
| | Fighting | A violent behavior displayed by each animal when locked together. Separate behaviors are difficult to distinguish properly due to the fast rolling over and over seen with the animals kicking, biting, and wrestling. The initial victim retaliates towards the attacker. Score for all mice actively involved in the fight. |
| Group Sleeping | | Sleeping that occurs when two or more mice are resting while in contact with the body of another mouse. When in the nest, the animals may not be seen clearly due to camera angles. Only score if the animals are observed going into and staying in a central resting area together once movement ceases for at least 5 seconds. This will typically be in the main nest, but they could remain behind bedding. |
| Allo-groom | | During grooming, the actor mouths and licks the fur on the recipient's body. The actor will also use its teeth to clean the hair shaft by pulling the fur from the base of the hair shaft upward or outward. |
| Social investigation | Face sniffing | A mouse sniffing the face of its cage mate |
| | Ano-genital sniffing | A mouse sniffing the ano-genital region of its cage mate |

**Nesting Behaviors- recorded every 30 minutes using one-zero sampling**

| Paw nesting | Digging | A series of at least 3 fast alternating movements of the forepaws scraping back material. The material will accumulate in a pile under the abdomen of the animal |
|---|---|---|
| | Push Dig | The forwards pushing and kicking of bedding material with fast alternating movements of the forepaws. It is accompanied by forward locomotion. |
| | Sorting- Paw | The placing of specific nesting or bedding material into a particular location, while sitting in the nest. Sorting is done in a deliberate fashion. |
| | Pulling In | The animal reaches out of the nest and pulls the nesting material in towards the nest. This may also be accomplished, by grasping the material in its mouth and dragging it in to the edge of the nest site. While performing this behavior the animal's hind legs do not leave the nest, and the forelegs are pulled back in each time the animal reaches out of the nest. |
| | Fluffing | This behavior can be unseen due to insufficient camera angles as it is characterized by the enlargement of the nest from the inside. The walls of the nest appear to jump as the whole nest enlarges. It is assumed that the inside of the nest is being hollowed out by the animal pushing the walls back and up. When visible, fast movement of the forepaws is seen as in push dig. However, no forward locomotion occurs while fluffing. |
| Oral nesting | Carrying | The animal is mobile while holding pieces of bedding or nesting material in its mouth. The material is transported to a new location in the cage. |
| | Sorting- Mouth | The placing of specific nesting or bedding material while sitting in the nest, done in a deliberate fashion using the mouth. Animal is not mobile as in "carrying" and does not chew the material is in "fraying". |
| | Fraying | The animal uses movement of the forepaws to draw material through the mouth. Gnawing movements of the jaw and jerking movements with the head are also seen. Score for oral manipulation/chewing of material. Do not score if the animal is chewing, but material pieces cannot be seen. |
| Active | | Score if the mouse is visible and moving for more than 5 seconds. |

cage, the scores from both nests were averaged. Daily values from each cage were used to determine the average score for the study week.

**Social ranking.** On days 5 and 6 of the study, the tube test was run to determine the linearity of each cage's social hierarchy based on Howerton et al. [51]. Previously, lower linearity has been reported with higher aggression levels [51]. The tube test was run over 2 days due to the time consuming nature of the pairwise tests for all mice within the cage. When conducting the test, strain was blocked by time of day to counteract systematic test order bias. That is, we tested one cage of every strain in each time period (morning (06:30–12:30) and afternoon (13:00–17:30)).

In brief, the test is conducted using a PVC tube (approx. 2.5cm diameter) connected to two plexiglass containers (approx. 19 cm x 19 cm x 21.5 cm). To acclimate the mice, 24 hours before the trials each mouse was placed in the test arena and given at least five, but no more than ten minutes to acclimate which was defined by the mouse comfortably exploring the areas on each side of the tube. Testing began by placing two mice from the same cage on opposite sides of the tube. They typically entered the tube immediately. The first mouse to place both hindfeet on the floor outside the tube was considered the loser. In a cage of five mice, there were ten different pairwise trials to test. All trials were repeated four times to give forty total trials per cage. The test arena was cleaned with ethanol and allowed to air dry between each trial. Trials were given a cutoff time of two minutes. Each mouse received a dominance score ($V_{ij}$) determined by the number of trials won by mouse $i$ when competing against mouse $j$. $V_{ij}$ scores were used to calculate the hierarchy linearity of the cage based on Landau's h [79].

$$h = \frac{12}{N^3 - N} \sum_{i=1}^{N} [V_i - \left(\frac{N-1}{2}\right)]^2$$

Where N = the number of mice per cage and $V_i$ is the summation of $V_{ij}$ for each mouse $i$ on its opponent mouse $j$. Scores near 1 correspond to a near complete hierarchy while scores near 0 signify the lack of a hierarchy. Each mouse's rank was also calculated by determining the number of trials won over all trials in which he participated. These scores were used to determine the dominant and subordinate mice used for sweat and saliva sampling.

## Statistical analysis

**Sample VOC profiles.** Before formal analysis, all VOC data were visualized using a Venn diagram to summarize similar and unique compounds across sample types. R Studio (version 3.4.3) and the *VennDiagram* package were used to create the visualization.

**Strain and VOC profiles.** Individual nest (N = 23), sweat (N = 46), and urine (N = 42) samples were separately visualized in two dimensions using non-metric multidimensional scaling (NMDS) to examine similarity based on VOC proportions across strain. Sweat and urine data were also examined for similarity between two levels of social rank. Factor differences were tested using the Adonis test since the datasets did not meet multivariate normality. Beta dispersion assumption was checked post hoc. Since cages were run in four groups over time, the batch number was also included as a blocking factor. NMDS, Adonis test, and assumption check were run in R Studio (version 3.4.3) using *vegan*, *tidyverse*, *ggplot2*, and *mvnormtest* packages.

Additionally, since mice were sampled based on ranking in the tube test, we wanted to confirm differences in two known urinary pheromones, β-farnesene and 2-sec-butyl-thiazoline (SBT) between social rank. Both pheromones have been previously reported to vary based on social rank [36]. Proportions of β-farnesene and SBT were analyzed using restricted maximum likelihood mixed models with strain, rank, and their interaction as fixed effects, and batch number as a random factor. Cage nested within strain was also included as a random factor to

account for repeated sampling from the same cage. The models were run in JMP Pro (version 14.0.0), and assumptions were checked post hoc.

**VOC profiles and social interactions.**　Cage level proportion data for each sample type (nest, sweat, urine) and social behavior were run in separate Principal Component Analyses (PCA) with values scaled to a mean of 0 and standard deviation of 1. The broken stick model (BSM) was used for principle component (PC) retention with the following exception: for the behavior PCA, BSM showed that only PC1 was significant. However, behavior PC2 explained a large portion of the variance, 29.67%, and had an eigenvalue of 1.48, therefore it was kept for further analysis. The following numbers in parentheses represent the number of retained PCs for each dataset: nest (2), sweat (1), urine (3), and behavior (2). Varimax rotation was used on the nest, sweat, and urine PCAs to maximize variable separation across PCs.

Mixed models were used to determine how nest, sweat, and urine odors affect behavior. Strain, and PCs from the nest, sweat, and urine data were used as independent variables, while PCs from the behavior data were tested separately as dependent variables. The cage average weekly nest score and Landau's H were included as covariates. Batch number was used a random factor. Non-significant variables were manually excluded from the models and those with the lowest AIC value were kept for interpretation. Since this study used two models to assess whether VOCs impact behavior, p values were adjusted using the sequential Bonferroni procedure to correct for multiple comparisons [37]. All further analyses examining strain effects on VOC PCs and individual VOCs were also run as mixed models. Individual VOC models had compound specific hypotheses and therefore a multiple comparisons correction was not performed. Individual VOCs were only tested in a mixed model if their PC of origin showed strong correlation with behavior based on Pearson's r. Normality and homogeneity of variance were tested post hoc by visually examining the residual Q-Q plot and spread of the residual by predicted plots for each model [80]. PCAs were run in R Studio (version 3.4.3) using *FactoMineR*, *factoextra*, and *tidyverse* packages. JMP Pro (version 14.0.0) was used for the mixed models and assumption check [80].

Data from one AJ nest was excluded due to flooding, making group sizes for the nest dataset unbalanced for NMDS, Adonis test, and PCA (AJ: n = 7, B6 and SJL: n = 8).

**Behavior across study days.**　To validate historical differences in strain social behavior and explore differences in strain nesting behavior, cage level behavior proportions from each day of observation (1, 2, 7) were tested in a series of REML mixed models with strain, day and the interaction as fixed effects, and batch number and cage nested within strain as random factors (N = 72, 3 observations from 24 cages). Post hoc Tukey tests were used to assess factor level differences. Assumptions of normality and homogeneity of variance were tested by visual examination of the residual Q-Q plot using JMP Pro (version 14.0.0) and Levene's test using R Studio (version 3.4.3) respectively. A $\log_{10}$ transformation was used on social investigation data, and square root transformations were used on the mediated aggression and allo-groom data. Data for escalated aggression was extremely skewed and transformation was unsuccessful to meet model assumptions. Therefore, count data per day were calculated and analyzed using a generalized linear mixed model (GLIM) with a negative binomial distribution. Custom tests corrected for multiple comparisons were used to identify specific factor differences.

## Supporting information

**S1 File. Methods for 3,5-diethyl- 2-hydroxycyclopent-2-en-1-one preparation and synthesis are submitted.**
(DOCX)

**S1 Fig.** (A) Total ion chromatogram (TIC); peak elutes at 31.562 minutes. (B) Full mass spectrum of 31.562 minute peak. (C) Molecular ion region. The boxes represent the theoretical distribution.
(TIF)

**S1 Table. Total number of data points used in each analysis.**
(XLSX)

**S2 Table. Effects of strain, social rank, and batch number on raw VOC proportions based on the Adonis test.**
(XLSX)

**S3 Table. Fixed effects on known urinary pheromones based on mixed models.**
(XLSX)

**S4 Table. Non-normalized nest GC-MS data.**
(XLSX)

**S5 Table. Calculated proportion nest GC-MS data (raw values/ total sample peak area).**
(XLSX)

**S6 Table. Non-normalized sweat GC-MS data.**
(XLSX)

**S7 Table. Calculated proportion sweat GC-MS data (raw values/ total sample peak area).**
(XLSX)

**S8 Table. Non-normalized urine GC-MS data.**
(XLSX)

**S9 Table. Calculated proportion urine GC-MS data (raw values/ total sample peak area).**
(XLSX)

**S10 Table. Non-normalized saliva GC-MS data.**
(XLSX)

**S11 Table. Legend matching raw data files in the Agilent ChemStation format to each respective cage or mouse sample ID.**
(XLSX)

**S12 Table. Behavior data across all study days.**
(XLSX)

**S13 Table. Behavior proportions from individual study days.**
(XLSX)

**S1 Data.**
(ZIP)

**S2 Data.**
(ZIP)

**S3 Data.**
(ZIP)

**S4 Data.**
(ZIP)

**S5 Data.**
(ZIP)

**S6 Data.**
(ZIP)

**S7 Data.**
(ZIP)

**S8 Data.**
(ZIP)

## Acknowledgments

All GC-MS data were acquired through the Institute for Pheromone Research at Indiana University in the Department of Chemistry. Data to verify 3,5-diethyl- 2-hydroxycyclopent-2-en-1-one structure were collected by Jonathan Karty using an Agilent 7890B/G7250 GC-QTOF-MS in the Indiana University Mass Spectrometry Facility. We'd like to thank our undergraduate assistants, Katie Bachert and Nicole Brockway, for the many hours spent coding video as well as Mikayla Burrell at the Institute for Pheromone Research for assisting with VOC sample analyses.

## Author Contributions

**Conceptualization:** Amanda J. Barabas, Jeffrey R. Lucas, Marisa A. Erasmus, Heng-Wei Cheng, Brianna N. Gaskill.

**Data curation:** Amanda J. Barabas, Helena A. Soini, Milos V. Novotny, David R. Williams, Jacob A. Desmond.

**Formal analysis:** Amanda J. Barabas, Brianna N. Gaskill.

**Funding acquisition:** Jeffrey R. Lucas, Marisa A. Erasmus, Heng-Wei Cheng, Brianna N. Gaskill.

**Investigation:** Amanda J. Barabas, Brianna N. Gaskill.

**Methodology:** Amanda J. Barabas, Helena A. Soini, Milos V. Novotny, David R. Williams, Jacob A. Desmond.

**Project administration:** Amanda J. Barabas.

**Resources:** Helena A. Soini, Milos V. Novotny, David R. Williams, Jacob A. Desmond.

**Software:** Helena A. Soini.

**Supervision:** Jeffrey R. Lucas, Marisa A. Erasmus, Heng-Wei Cheng, Brianna N. Gaskill.

**Writing – original draft:** Amanda J. Barabas.

**Writing – review & editing:** Helena A. Soini, Milos V. Novotny, David R. Williams, Jacob A. Desmond, Jeffrey R. Lucas, Marisa A. Erasmus, Heng-Wei Cheng, Brianna N. Gaskill.

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
