## [Decision Letter · Decision Letter 0]

22 Sep 2020

PONE-D-20-24539

Friendship stinks: Plantar foot sweat, nesting material, and urine contain compounds associated with affiliative behaviors in group housed male mice, Mus musculus

PLOS ONE

Dear Dr. Barabas,

Thank you for submitting your manuscript to PLOS ONE. After careful consideration, we feel that it has merit but does not fully meet PLOS ONE’s publication criteria as it currently stands. Therefore, we invite you to submit a revised version of the manuscript that addresses the points raised during the review process.

Based on the comments of experts in the field, the manuscript presents several major weaknesses. Among these are the correlative nature of the study , the assumption about the nature of molecules involved in the communication , the statistical approach. A profound revision is needed to make the work exploitable by readers.

We look forward to receiving your revised manuscript.

Kind regards,

Igor Branchi, Ph.D.

Academic Editor

PLOS ONE

Journal Requirements:

Reviewers' comments:

Reviewer's Responses to Questions

**Comments to the Author**

1. Is the manuscript technically sound, and do the data support the conclusions?

Reviewer #1: Partly

Reviewer #2: Partly

2. Has the statistical analysis been performed appropriately and rigorously? 

Reviewer #1: I Don't Know

Reviewer #2: No

3. Have the authors made all data underlying the findings in their manuscript fully available?

Reviewer #1: Yes

Reviewer #2: No

4. Is the manuscript presented in an intelligible fashion and written in standard English?

Reviewer #1: Yes

Reviewer #2: Yes

5. Review Comments to the Author

Reviewer #1: In this study, Barabas and colleagues collected body odours (urine, saliva, and plantar sweat, as well as nest odours collected from nesting material) from 3 mouse strains, in an attempt to provide a factual basis to the empirical observation that reusing bedding material reduces aggression (through olfactory signals) upon cage changing in mouse husbandry. Compounds were identified by GCMS and their abundance correlated to various behaviours (mostly pertaining to social/aggressive interactions). Some odour profiles were correlated with affiliative behaviours. Please note, the manuscript would have been clearer if the figure legends had featured together out of the main text body.

Improvement of lab animal welfare is hugely important and it is important to promote practices that lead to happier animals, for ethical and scientific reasons (to limit the effect of stress on the physiology we study). But one could ask: if reusing soiled litter has been sufficiently shown to work (ref 12), do we need to know why? I am being provocatively pragmatic, but perhaps this point should be argued in the paper.

This study is well written and there are some very positive and interesting elements (e.g. the GC-MS analysis of cardboard mixed with bedding material is a great idea, for the study of body odours in nests). I also warmly welcome the fact that the authors have openly been forthright and discussed potential problems/limitations (e.g. use of pilocarpine).

My main issue is that the study is merely correlative and based on a number of assumptions. The authors have not claimed otherwise and this is reflected in the title (‘contain compounds associated with affiliative behaviors’) and rest of the manuscript. But what to make of this data if we don’t compare the efficiency of the soiled bedding material (and the olfactory signals it contains) in preventing aggression between cagemates? Perhaps some of strains have different olfactory performance? Instead of the putative pheromones, these molecules could simply be metabolites reflecting the product of differences in physiology and microbiota between the 3 strains tested. We cannot (and the authors have exerted restraint in their write up) assume any causality between the VOC signatures and behaviours. This lack of functional data severally limits the message/data interpretation.

Was this the best way to test the hypotheses laid out in the introduction? I cannot help thinking that if I had wanted to address those questions, I would have envisaged a study where soiled nesting material would be used to investigate modulation in aggression/affiliative behaviour. And then tested the odour signature. Otherwise the correlations are not grounded in any factual reality.

Abstract: ‘This supports nest transfer as a recommended practice during cage change.’ … I am not sure this is right, how do the correlations described here support the practice??

Title: I do love a joke and I am in favour in making scientific literature more exciting, but I am afraid I am not sure ‘friendship stinks’ contributes to a good informative title for a scientific article (in fact might it be slightly misleading?). I would just point out that (a) there is no indication these odours smell bad (perhaps they might even be attractive) and (b) the evidence between these odour and affiliative behaviours is merely correlative.

Introduction: it is a matter of opinion, but the mention of pheromone insinuates that the compounds identified will have pheromonal properties and causally affect behaviour (and this study does not show this). Even if the odours identified reduced aggression, this study does not show the compounds are pheromones, they could just be familiar odours.

- Dominance: any anecdotal observation that the tube test correlated to other measures (e.g. small wounds, scruffy appearance)?

- Role of pilocardine: Is there any evidence/data/comparison of VOC profiles with non pilocardine treated individuals? (even in other types of body fluids) are there individual differences in pilocarpine response (ie differences in physiology linked to social rank/anxiety status etc)

- Plantar sweat (line 69 and throughout) : I wonder whether the secretions were specifically that of a specific gland (producing that oily substance mentioned) or the product of all sweat glands on the injected foot area.

- I am not sure the risk of contamination with fecal odours has been adequately discussed.

- Affiliative behaviour: Since this is central to this work, please ensure you provide a clear definition of affiliative behaviour (in the intro) and of the behaviours that belong to this category. I found examples in Fig 7 legend, but I believe this should be made clearer in the text.

Comments:

Table 1: browsing through the compounds detected, I found a couple of surprises, I checked that methyldihydrojasmonate has also been detected in human saliva (DOI: 10.1007/s10886-010-9846-7), however can the authors explain why limonene can be detected in mouse saliva?

Line 61: the statement ‘In contrast, little is known about pheromones that may promote affiliative behaviors among male mice’. Is this compatible with line 65 ‘Even though mice are not known to produce affiliative odors’?

Line 76: should it be ‘aim’ (singular)?

Line 83-85 (working hypothesis): this was not tested.

Line 91: to be more precise, add ‘cage per strain’ after n=8?

Line 317-318: it has been reported that tube test hierarchy assessment after one measure does not provide a reliable indicator (doi: 10.1038/s41598-018-24624-4)

Line 358 : missing punctuation

Line 366: the discussion on the consequences of pilocarpine use is very welcome.

Line 470: pathogens/pathogen agents instead of agent?

Line 472-473: traditional control? The inter strain comparison does not require a control (what would it be?!?), what does this have to do with being an exploratory study? Consider rephrasing.

Line 486: I assume mentioning experimenter’s sex is in reference to studies demonstrating the impact on lab rodents (e.g. Sorge et al 2014). Should the rationale/reference be mentioned?

Line 495: partially sighted or blinded to treatment?

Reviewer #2: Previous studies have shown that transferring soiled nest material when groups of male mice are transferred into a clean cage can reduce their aggression. The main aim of this study was to quantify volatile organic compounds (VOCs) in nest material soiled by male mice, identify the likely sources of nest odor compounds (from urine, saliva or plantar glands), and establish whether differences in aggression levels between strains correlate with levels of different VOCs (or combinations). A second aim was to determine the relationship between VOC profiles and mouse social behavior (more specifically, aggressive and affiliative behavior), particularly to test the hypothesis that VOCs in soiled nest material and plantar gland secretion (here referred to as sweat) reduce aggression and promote positive (affiliative) social interactions.

The manuscript contains some potentially important and novel data, particularly providing the first chemical analysis of nest material soiled by male mice and their plantar gland secretions. However, there are some fundamental issues with the experimental design and analysis that need to be addressed before it is possible to review the findings and conclusions of the study. As such, my review focuses only on these issues, together with a suggestion for improving the way that the findings are presented to make the study much more accessible to readers.

The study design uses eight replicate cage groups of 5 males for each of three inbred strains, selected due to differences in aggressiveness and because B6 is the most commonly used inbred mouse strain. Analyses confirm that the three strains showed highly significant differences in behavior and in VOC profiles for all three odor sources (nest, sweat and urine). Subsequent analyses then attempt to look at the relationships between VOC profiles for each odor source per cage and group social interactions, first using principal components analyses to reduce the number of variables and take into account correlation between individual VOCs or behaviors. However, to analyse correlations, the data are pooled across strains and strain is not included in the analyses. This is inappropriate – the replicates are not independent and drawn randomly from a population of mice, but instead represent replication of three independent genotypes. As strains were picked a priori for differences in social behavior, and (from previous work) differences in VOC profiles are expected between genetically distinct strains, correlations could simply be due to strain differences without implying any direct relationship between behavior and VOCs (just as genetically distinct strains differ in many other characteristics). Including strain in the analysis will address whether variation in VOC profiles between cages relates to differences in aggression or affiliative behaviour within and across genotypes. Currently, it is not possible to see how much affect this might have on analyses as strain is not shown in figures showing correlations (Fig 3, 4), but this is essential information. There are a number of ways to achieve this, but one possibility might be to use mixed effect models, which would also allow block to be included as a random factor (as in other analyses) and allows for some missing data without have to exclude all data for a particular cage (but this is just a suggestion). As the study is looking for any differences in VOC PCs that relate to either of two different measures of behavior, p values also need to be corrected for the multiple comparisons that essentially all address the same question (is there a relationship between behavior and VOC profile?), but I couldn’t see any mention of this in the statistical analysis section.

It is unclear in places how repeated measurements have been taken into account in analyses (ie where more than one data point was used per cage). For example, for the analysis of nesting behavior and other behaviors across the week (lines 255-268) the n size is given as 72 or 67 degrees of freedom for the error term, but this is surely incorrect. The n size was 24 cages, with repeated measures on 3 days. Consider using a repeated measures GLM with day as a within-subjects factor.

I would also strongly encourage the authors to rethink the way that results are presented so that the findings are much more accessible to a reader – while this study is within my area of expertise and fundamental interest, I found the multiple statistical analyses presented in the text to be impenetrable and impossible to follow without drawing out my own summary tables. It would be a massive shame to put all this effort into the study but then have very few readers able to read and understand it. Could I suggest that analyses are provided in Tables so that the text focuses on summarising the main patterns of relationships from the perspective of the key question being addressed here – whether behavioural variation in aggression or in affiliation between groups is significantly related to VOC profile of nest material and/or specific odor sources, and which components might then be of particular importance. A diagram summarising key strain differences might also help. Given that the focus of the study is on the odors in nest material that may reduce aggression between males, this gets lost in the Discussion section. I would suggest that the authors make sure that this is the focus and include an appropriate critique of what conclusions can be drawn about the influence of VOCs on behavior from the patterns of correlation found (particularly across a small number of strains) and what further evidence is needed. Other points could then be addressed secondarily.

Please note that the colors used to distinguish between strains in Fig. 2 are difficult to distinguish, particularly when printed out.

Data abstracted from GC/MS analyses are included in supplementary material but it is not clear whether the raw data files on which these values are based have been submitted to a depository. I could not find reference to the availability of raw behavioral data but this will be necessary to meet the PLOS Data policy.

I hope these comments are helpful.

6. PLOS authors have the option to publish the peer review history of their article (what does this mean?). If published, this will include your full peer review and any attached files.

Reviewer #1: No

Reviewer #2: No

---

## [Author Response · Author response to Decision Letter 0]

2 Dec 2020

Note: This response is submitted as a word document and formatted to distinguish reviewer comments from author responses. 

We humbly thank the reviewers for the comments below. We feel the suggested edits improve the both the scientific quality and reader comprehension of this manuscript. Each individual point has been addressed below. 

Reviewer #1: 

In this study, Barabas and colleagues collected body odours (urine, saliva, and plantar sweat, as well as nest odours collected from nesting material) from 3 mouse strains, in an attempt to provide a factual basis to the empirical observation that reusing bedding material reduces aggression (through olfactory signals) upon cage changing in mouse husbandry. Compounds were identified by GCMS and their abundance correlated to various behaviours (mostly pertaining to social/aggressive interactions). Some odour profiles were correlated with affiliative behaviours. Please note, the manuscript would have been clearer if the figure legends had featured together out of the main text body.

Thank you for the feedback. We followed the submission instructions of the author guide so we cannot alter how legends and figures are presented in the main text body.

Improvement of lab animal welfare is hugely important and it is important to promote practices that lead to happier animals, for ethical and scientific reasons (to limit the effect of stress on the physiology we study). But one could ask: if reusing soiled litter has been sufficiently shown to work (ref 12), do we need to know why? I am being provocatively pragmatic, but perhaps this point should be argued in the paper.

As suggested, further justification has been added in the introduction, see lines 85-89 (clean version). 

This study is well written and there are some very positive and interesting elements (e.g. the GC-MS analysis of cardboard mixed with bedding material is a great idea, for the study of body odours in nests). I also warmly welcome the fact that the authors have openly been forthright and discussed potential problems/limitations (e.g. use of pilocarpine).

My main issue is that the study is merely correlative and based on a number of assumptions. The authors have not claimed otherwise and this is reflected in the title (‘contain compounds associated with affiliative behaviors’) and rest of the manuscript. But what to make of this data if we don’t compare the efficiency of the soiled bedding material (and the olfactory signals it contains) in preventing aggression between cagemates? Perhaps some of strains have different olfactory performance? Instead of the putative pheromones, these molecules could simply be metabolites reflecting the product of differences in physiology and microbiota between the 3 strains tested. We cannot (and the authors have exerted restraint in their write up) assume any causality between the VOC signatures and behaviours. This lack of functional data severally limits the message/data interpretation.

Was this the best way to test the hypotheses laid out in the introduction? I cannot help thinking that if I had wanted to address those questions, I would have envisaged a study where soiled nesting material would be used to investigate modulation in aggression/affiliative behaviour. And then tested the odour signature. Otherwise the correlations are not grounded in any factual reality.

Yes, it is possible that differences in strain compounds could be due to metabolic, microbiome, and/or unidentified genetic differences. However, the intent of this study was to serve as the first step of a framework laid out to identify pheromones in plantar sweat (Wyatt T. Semiochemicals: Pheromones, Signature Mixtures and Behaviour. In: Nielsen BL, editor. Olfaction in Animal Behaviour and Welfare. Boston, MA: CABI; 2017. pp. 36–38.) and whether there are strain differences in the types of compounds produced; the intent was not to examine whether soiled nesting material reduces aggression. Before a casual effect is tested, it is first necessary to examine what types of compounds are present in nesting material and then to examine whether there is a quantifiable behavior difference associated with a key molecule(s). This rationale has been added to the end of the introduction to further clarify the goal of the current project (lines 105-110, clean version ). 

Abstract: ‘This supports nest transfer as a recommended practice during cage change.’ … I am not sure this is right, how do the correlations described here support the practice??

Conclusion statement has been rephrased.

Title: I do love a joke and I am in favour in making scientific literature more exciting, but I am afraid I am not sure ‘friendship stinks’ contributes to a good informative title for a scientific article (in fact might it be slightly misleading?). I would just point out that (a) there is no indication these odours smell bad (perhaps they might even be attractive) and (b) the evidence between these odour and affiliative behaviours is merely correlative.

We have changed the manuscript title. 

Introduction: it is a matter of opinion, but the mention of pheromone insinuates that the compounds identified will have pheromonal properties and causally affect behaviour (and this study does not show this). Even if the odours identified reduced aggression, this study does not show the compounds are pheromones, they could just be familiar odours.

We did not intend to imply that what we were measuring were in fact, pheromones. Pheromones were only mentioned in order to give an overview of olfactory communication, and as an example of what may be held in nesting material. This section has been clarified. 

- Dominance: any anecdotal observation that the tube test correlated to other measures (e.g. small wounds, scruffy appearance)?

Anecdotally, SJL mice with less wounding seemed to perform worse in the tube test. However, we did not formally test this so have not included this in the discussion.

- Role of pilocarpine: Is there any evidence/data/comparison of VOC profiles with non pilocarpine treated individuals? (even in other types of body fluids) are there individual differences in pilocarpine response (ie differences in physiology linked to social rank/anxiety status etc)

From what we could find, the only VOC profile comparison was done in human subjects either subjected to pilocarpine, or exercise to promote excess sweating. Whether this is comparable to mice is debatable, since mice do not sweat in the same way humans do to thermoregulate, but this has been added to the discussion. There is also documentation of individual variation in pilocarpine success for treating dry mouth, but factors behind this variation do not seem to be known. This has also been added as well as the volume range of saliva samples in this study. 

- Plantar sweat (line 69 and throughout) : I wonder whether the secretions were specifically that of a specific gland (producing that oily substance mentioned) or the product of all sweat glands on the injected foot area.

The secretions from mouse foot pads are known to specifically come from eccrine sweat glands. This has been added to the text.

- I am not sure the risk of contamination with fecal odours has been adequately discussed.

It was mentioned in the discussion that fecal odors could be present in the nest material (line 310, clean version). However, we are unsure if this comment references specifically the contamination of nest material. Since we did not measure fecal odors in these samples, we do not feel comfortable speculating their role on the behavior observed here. 

To avoid contamination with fecal odors, the methods describe how the surface of the foot and the anesthesia chamber were cleaned with ethanol before sweat and saliva were collected, respectively. Urine was also collected from surfaces that either never held feces or were cleaned with ethanol before the mice were placed on them. 

- Affiliative behaviour: Since this is central to this work, please ensure you provide a clear definition of affiliative behaviour (in the intro) and of the behaviours that belong to this category. I found examples in Fig 7 legend, but I believe this should be made clearer in the text.

A definition and examples of affiliative behavior have been added to the introduction (line 66-68, clean version).

Comments:

Table 1: browsing through the compounds detected, I found a couple of surprises, I checked that methyldihydrojasmonate has also been detected in human saliva (DOI: 10.1007/s10886-010-9846-7), however can the explain why limonene can be detected in mouse saliva?

Limonene is a common mammalian metabolite, and it has been detected previously in human saliva (Huda AL-Kateb et al 2013 J. Breath Res. 7 036004). The methods do vary in these human papers, so that likely influences the chance of detection. However, the methods in our study are based on the first, where limonene wasn’t found. Perhaps it also depends on geographic region and local microbiome as limonene is produced by microorganisms as well (https://doi.org/10.1080/10408440802291497; Table 2)

Line 61: the statement ‘In contrast, little is known about pheromones that may promote affiliative behaviors among male mice’. Is this compatible with line 65 ‘Even though mice are not known to produce affiliative odors’?

The latter has been reworded. 

Line 76: should it be ‘aim’ (singular)?

This typo has been fixed.

Line 83-85 (working hypothesis): this was not tested.

This line has been edited to not assume causality. 

Line 91: to be more precise, add ‘cage per strain’ after n=8?

Added

Line 317-318: it has been reported that tube test hierarchy assessment after one measure does not provide a reliable indicator (doi: 10.1038/s41598-018-24624-4)

Unfortunately, the referenced study was published while we were running this project. The citation is in the manuscript and we acknowledge that our tube test protocol may not have been the best method.

Line 358 : missing punctuation

Added

Line 366: the discussion on the consequences of pilocarpine use is very welcome.

Thank you for this comment. 

Line 470: pathogens/pathogen agents instead of agent?

Edited

Line 472-473: traditional control? The inter strain comparison does not require a control (what would it be?!?), what does this have to do with being an exploratory study? Consider rephrasing.

Line 486: I assume mentioning experimenter’s sex is in reference to studies demonstrating the impact on lab rodents (e.g. Sorge et al 2014). Should the rationale/reference be mentioned?

This manuscript participated in beta testing for the ARRIVE 2.0 guideline. Both points were recommended for inclusion to improve the consistency of reporting. Regardless, a clarification to the statement has been added. 

Line 495: partially sighted or blinded to treatment?

Edited 

Reviewer #2: 

Previous studies have shown that transferring soiled nest material when groups of male mice are transferred into a clean cage can reduce their aggression. The main aim of this study was to quantify volatile organic compounds (VOCs) in nest material soiled by male mice, identify the likely sources of nest odor compounds (from urine, saliva or plantar glands), and establish whether differences in aggression levels between strains correlate with levels of different VOCs (or combinations). A second aim was to determine the relationship between VOC profiles and mouse social behavior (more specifically, aggressive and affiliative behavior), particularly to test the hypothesis that VOCs in soiled nest material and plantar gland secretion (here referred to as sweat) reduce aggression and promote positive (affiliative) social interactions.

The manuscript contains some potentially important and novel data, particularly providing the first chemical analysis of nest material soiled by male mice and their plantar gland secretions. However, there are some fundamental issues with the experimental design and analysis that need to be addressed before it is possible to review the findings and conclusions of the study. As such, my review focuses only on these issues, together with a suggestion for improving the way that the findings are presented to make the study much more accessible to readers.

The study design uses eight replicate cage groups of 5 males for each of three inbred strains, selected due to differences in aggressiveness and because B6 is the most commonly used inbred mouse strain. Analyses confirm that the three strains showed highly significant differences in behavior and in VOC profiles for all three odor sources (nest, sweat and urine). Subsequent analyses then attempt to look at the relationships between VOC profiles for each odor source per cage and group social interactions, first using principal components analyses to reduce the number of variables and take into account correlation between individual VOCs or behaviors. However, to analyse correlations, the data are pooled across strains and strain is not included in the analyses. This is inappropriate – the replicates are not independent and drawn randomly from a population of mice, but instead represent replication of three independent genotypes. As strains were picked a priori for differences in social behavior, and (from previous work) differences in VOC profiles are expected between genetically distinct strains, correlations could simply be due to strain differences without implying any direct relationship between behavior and VOCs (just as genetically distinct strains differ in many other characteristics). Including strain in the analysis will address whether variation in VOC profiles between cages relates to differences in aggression or affiliative behaviour within and across genotypes. Currently, it is not possible to see how much affect this might have on analyses as strain is not shown in figures showing correlations (Fig 3, 4), but this is essential information. There are a number of ways to achieve this, but one possibility might be to use mixed effect models, which would also allow block to be included as a random factor (as in other analyses) and allows for some missing data without have to exclude all data for a particular cage (but this is just a suggestion). As the study is looking for any differences in VOC PCs that relate to either of two different measures of behavior, p values also need to be corrected for the multiple comparisons that essentially all address the same question (is there a relationship between behavior and VOC profile?), but I couldn’t see any mention of this in the statistical analysis section.

Thank you for these comments. The analyses have been redone to include strain in the main PC models and mouse batch number (block). Correction for multiple comparisons has been done using the Bonferroni sequential method which is particularly useful for “areas that are in their infancy or those that are underexplored ,” (DOI:10.3233/NRE-130893).

It is unclear in places how repeated measurements have been taken into account in analyses (ie where more than one data point was used per cage). For example, for the analysis of nesting behavior and other behaviors across the week (lines 255-268) the n size is given as 72 or 67 degrees of freedom for the error term, but this is surely incorrect. The n size was 24 cages, with repeated measures on 3 days. Consider using a repeated measures GLM with day as a within-subjects factor.

In the models that analyze behavior for each study day, cage, nested in strain, is included as random factor. We did have an oversight and originally forgot to include it in the two nesting models, which has been corrected. 

I would also strongly encourage the authors to rethink the way that results are presented so that the findings are much more accessible to a reader – while this study is within my area of expertise and fundamental interest, I found the multiple statistical analyses presented in the text to be impenetrable and impossible to follow without drawing out my own summary tables. It would be a massive shame to put all this effort into the study but then have very few readers able to read and understand it. Could I suggest that analyses are provided in Tables so that the text focuses on summarising the main patterns of relationships from the perspective of the key question being addressed here – whether behavioural variation in aggression or in affiliation between groups is significantly related to VOC profile of nest material and/or specific odor sources, and which components might then be of particular importance. A diagram summarising key strain differences might also help. 

Thank you for this comment. Several tables have been added to the main text and supplementary data to present statistical values. We also included a strain summary diagram at the end of the results. We appreciate this feedback to help make our manuscript easy to understand. 

Given that the focus of the study is on the odors in nest material that may reduce aggression between males, this gets lost in the Discussion section. I would suggest that the authors make sure that this is the focus and include an appropriate critique of what conclusions can be drawn about the influence of VOCs on behavior from the patterns of correlation found (particularly across a small number of strains) and what further evidence is needed. Other points could then be addressed secondarily.

The discussion has been reorganized in an attempt to address this comment. 

Please note that the colors used to distinguish between strains in Fig. 2 are difficult to distinguish, particularly when printed out.

Colors have been adjusted. 

Data abstracted from GC/MS analyses are included in supplementary material but it is not clear whether the raw data files on which these values are based have been submitted to a depository. I could not find reference to the availability of raw behavioral data but this will be necessary to meet the PLOS Data policy.

Thank you for pointing this out. I misread the requirement and raw GCMS files are submitted as compressed zip files in the Supplementary Information. However, they were obtained with a proprietary software and can only be opened with an Agilent GC-MS instrument with ChemStation software. 

I hope these comments are helpful.

Yes, very much so!

---

## [Decision Letter · Decision Letter 1]

4 Feb 2021

PONE-D-20-24539R1

Compounds from plantar foot sweat, nesting material, and urine show strain patterns associated with affiliative behaviors in group housed male mice, Mus musculus

PLOS ONE

Dear Dr. Barabas,

Thank you for submitting your manuscript to PLOS ONE. After careful consideration, we feel that it has merit but does not fully meet PLOS ONE’s publication criteria as it currently stands. Therefore, we invite you to submit a revised version of the manuscript that addresses the points raised during the review process.

The reviewers appreciated the improvements made by the authors. However, further changes are requested. In particular, strains differences should be better illustrated and considered in the interpretation of the results. 

We look forward to receiving your revised manuscript.

Kind regards,

Igor Branchi, Ph.D.

Academic Editor

PLOS ONE

Reviewers' comments:

Reviewer's Responses to Questions

**Comments to the Author**

1. If the authors have adequately addressed your comments raised in a previous round of review and you feel that this manuscript is now acceptable for publication, you may indicate that here to bypass the “Comments to the Author” section, enter your conflict of interest statement in the “Confidential to Editor” section, and submit your "Accept" recommendation.

Reviewer #1: All comments have been addressed

Reviewer #2: (No Response)

2. Is the manuscript technically sound, and do the data support the conclusions?

Reviewer #1: Yes

Reviewer #2: Partly

3. Has the statistical analysis been performed appropriately and rigorously? 

Reviewer #1: Yes

Reviewer #2: Yes

4. Have the authors made all data underlying the findings in their manuscript fully available?

Reviewer #1: Yes

Reviewer #2: Yes

5. Is the manuscript presented in an intelligible fashion and written in standard English?

Reviewer #1: Yes

Reviewer #2: Yes

6. Review Comments to the Author

Reviewer #1: I thank and congratulate the authors for their work to improve this manuscript. This report is very interesting and I am looking forward to reading up on future developments on the question of mouse aggression and plantar sweat. I would however suggest not to forget other odorant sources (e.g. urine or saliva), volatile compounds evaporate and can evade detection, and rarely biologically work in isolation (ie they are naturally occurring as part of mixtures, even though the literature glorifies single molecules like PEA, TMT and the likes).

Comments on ‘clean version’,

- Line 594 : is there a typo in this sentence?

- Line 635: were utilized?

- Throughout: I am not sure p’s is the plural of p, yet it could be this inappropriate use of apostrophe might be the best abbreviation for ‘p-values’

Reviewer #2: I am happy that the authors have responded appropriately to my major concern that strain was not included in correlation analyses between VOCs from different sources and behaviors, and that sequential Bonferroni has now been applied to adjust p values for multiple comparisons. An oversight to include day where appropriate as a within-subjects factor to avoid pseudoreplication has been corrected. I also found that inclusion of the new Tables 3 and 4 was very helpful in showing the patterns found in the data analysis much more clearly, while new Figure 6 provides a useful summary from such a complex data analysis (though I didn’t understand the color coding, see also other comments on the figure below). My review therefore focuses on interpretation of the data analysis and conclusions drawn. Some specific suggestions for some rewording are provided at the end of this review.

The focus of the study is on identifying candidate VOCs in nest material that might account for reduced aggression observed previously when soiled nest material is transferred between cages at cage cleaning. VOCs associated with affiliative behavior (allogrooming and group sleep) are also examined, although relevance to the reduction in aggression in response to soiled nest material isn’t very clear, given aggression and affiliative behavior do not show a simple negative relationship (as confirmed in this study). As only affiliative but not aggressive behavior features in the title and much of the conclusions, this needs some further explanation as I struggled to understand the relevance to the main issue addressed here.

The study is designed to look for VOC candidates that might have common effects in regulating aggression across mouse strains. I think it is important to clarify this underlying assumption, as differences between chosen strains in the odors used to regulate aggression, or in sensitivity to these VOCs, would result in a lack of association using this design. An example is mentioned in the Discussion (though not in this context) where strains differ in expression of a scent protein that stimulates aggression (lines 445-450), but nonetheless strain differences in expression do not correspond to differences in aggression but to strain lineage, which deserves further comment.

Although not anticipated when selecting the three strains used, the study found that aggression within two of the strains was rare during the week-long study and common only in one of the strains (SJL). This is important information for interpretation of the findings but this only becomes apparent to the reader quite late in the Results section, from Figure 5. PCA analysis of behavior derived two components that distinguish the main strain differences but it is important to refer to the data in Figure 5 to interpret these components given the major strain differences – currently the analysis is quite misleading (though I’m sure this is not intentionally so). Scores on PC1 (referred to as Aggression PC) are influenced not only by aggression and social investigation but also by high negative weighting to allo-grooming (not shaded in Table 2 but this has just as strong a weighting as others that are shaded). The highly significant difference between strains reported for this component (SJL > AJ > B6) is not because AJ has greater aggression than B6, which the text implies (Table 3, lines 191-192). Instead, the weightings on this component reflect greater aggression and social investigation in SJL compared to AJ and B6, but greater allogrooming in B6 than AJ or SJL (see Figure 5, leading to SJL > AJ > B6). This component thus discriminates the major differences in behavior between strains, but cannot be interpreted as reflecting aggression alone and the label Aggression PC is quite misleading. PC1 does not entirely explain behavioral differences between strains because group sleep shows a slightly different pattern (but accounts for less variance). The second component (referred to as Affiliative PC) reflects this strain difference in group sleep while being orthogonal to PC1 (which has already accounted for strain differences in aggression, investigation and allogrooming). Thus PC2 weightings reflect greater group sleep in B6 and SJL vs AJ while AJ also has lower allogrooming than B6 and lower aggression and social investigation than SJL so scores very low for this component. PCA merely finds the main axes of variation in the data – interpretation of the derived components and strain differences in scores requires inspection of the underlying behaviors shown in Figure 5, so it does not make sense to me to split up the analysis of behavior using PCA and the later section on strain and behavior.

Once the pattern of differences between strains is clear, one can then relate this in an appropriate way to VOC differences. As it stands, the implication of Table 3 and text is that aggression follows the pattern SJL > AJ > B6, so VOCs matching differences in aggression should be either SJL > AJ > B6 or SJL < AJ < B6. Instead, it is clear from Figure 5 that VOCs matching differences in aggression between strains should discriminate SJL from the other two strains and that “Aggression PC” is misleading. With that in mind, nest PC1 scores differ between SJL and the other two strains. However, as data on levels of the main compounds that contribute to nest PC1 are not presented, we cannot see which components differ consistently between SJL and both other strains (consistent with the hypothesis that some VOC components match strain differences in aggression) – important as the difference here is only just statistically significant while the strain difference in behavior is substantial. Would it not be better to focus detailed descriptions on this specific nest PC1 component as the one that does correspond with the strain difference of interest while also relating to nest material known to have aggression reducing properties, rather than losing the key relationship here in all the detail of VOC profiles that do not match the hypothesis being tested? The behavioral analysis clearly shows that the pattern of neither allogrooming nor group sleep (referred to as affiliative behaviors) correspond to strain differences in aggression, which needs to be pointed out much more clearly since the point of discussing VOC patterns in relation to ‘affiliative behavior’ (which do not correlate across strains and cannot simply be grouped together) is very unclear.

I do think there are interesting and novel data here, but this is being lost in an over-complex analysis that seems to be leading to confusion rather than insight. I would really advise the authors to consider focusing first on the strain differences in behavior which are quite clear cut. The finding that SJL was the only strain to show appreciable aggression then changes the focus of analysis of VOC profiles and could be presented in a very straightforward way. Allogrooming and group sleep each show different strain patterns so it is not possible to simply group these together as ‘affiliative behaviors’ in relation to strain differences from these analyses, and authors need to avoid the implication that mice that are affiliative are therefore not aggressive. If the main focus of the study is on VOCs that might appease aggression, shouldn’t the title reflect the study findings in relation to aggression not in relation to uncorrelated affiliative behaviors? It is just a suggestion, but throughout I found this confusing.

The discussion is extremely long and interpretations throughout really need revisiting in light of the points made above concerning the interpretation of behavior PCs and their match to VOCs. As a suggestion, perhaps some of the more speculative aspects could be dropped so that the focus remains clearly on the findings of the study.

Some specific points, particularly in relation to conclusions from analyses

Abstract line 33: The differences according to strain do not ‘mask direct effects on behavior’ in these analyses. Including strain in analyses increased the likelihood of seeing overall correlations between individual VOC profiles and behaviors, by taking into account any differences in levels of behavior or VOC profiles between strains. Instead analyses showed that there were no consistent relationships between individual VOC profiles and behavior that were not accounted for by major strain differences.

Abstract lines 34-37: As the data show that the two affiliative behaviors analysed were not correlated (thus would be better referred to separately), while differences in these behaviors between strains did not match differences in aggression, this seems to miss the main results and conclusions that can be drawn from the data.

Abstract line 37: The term ‘correlation’ is not appropriate here as this implies a continuous statistical relationship which was not shown with 3 strains. Strain differences in VOC patterns that match strain differences in behavior would be more appropriate as a conclusion, being appropriately cautious here as there is no statistical validity in just picking out the VOC profiles that match.

Abstract line 42: ‘provide preliminary information about a potential connection between …..

Intro line 66-68: As affiliative behavior is often not negatively correlated with aggression (more social animals generally show more of both), the relevance of affiliative behavior to the issue of appeasing odors in nest material needs more explanation. Does addition of soiled nest material result in increased affiliative behaviors as well as reduced aggression?

Intro line 81: This sentence doesn’t seem to make sense. Do you mean the mechanism behind the reduction in aggression observed in response to used nest material?

Intro lines 91-104: There needs to be some introduction to the choice of strains included in the study here, particularly when the next section is Results rather than Methods. Readers may not be familiar with these strains and what is already known about their aggressive / affiliative behavior.

Intro, last sentence: This would be more suitable for Discussion rather than introduction to this study.

Results (specific comments, see main comments above about strain differences and interpretation of PCA components)

Line 128: I would have found ‘Sample VOC profiles’ more helpful here (everything is a sample)

Line 133-134: I couldn’t see any indication in Table 1 for which compounds had been previously identified as mouse urinary compounds.

Line 173: I would avoid the term ‘significant’ here as it implies statistical significance relative to random.

Line 176: See main comments – I think giving these interpretive terms to the behavior PCs is misleading given the actual strain differences in the behaviors contributing to these components.

Line 178: Again, there is no validity to using ‘significant’ here perhaps ‘considered important’. As indicated above, this really requires inspection of the individual component data for interpretation but this hasn’t been included.

Table 2: ‘mediated’ and ‘escalated’ mean nothing to the reader – there is space in the Behavior column to include ‘aggression’.

Table 4: Given the importance of strain differences in these different behaviors for any interpretation, could you indicate in the table which strains are significantly different from which (as in Table 3? As none of the strain x day interactions are statistically significant, this could be simply stated in a footnote, giving appropriate space for showing the pattern of strain differences. As indicated in main comments, I think this section needs to come much earlier so that the PCA analyses and behavioral match to VOC patterns are interpretable. Given that aggression was so much stronger in SJL than in the other two strains, it would make more sense to match VOC patterns to this difference than to principal components that represent a much more complex pattern of differences between strains.

Line 277: In summary, SJL displayed substantially more aggressive behavior?

Line 279 & Figure 6: B6 mice did not display more group sleep than SJL

Discussion

Line 292: As indicated, Aggression PC does not only include aggression, but your data do show that only SJL showed appreciable aggression.

Line 299: Some of the component weights in nest PC1 are negative (low levels would result in high scores)

Lines 297- 301, 315-330: Does the pattern of urine PC2 scores really match the differences in aggression (not shown – you could plot this)?

Line 334-335: Surely sniffing behavior denotes gaining olfactory information. I don’t think anyone working in olfactory communication would define this as ‘affiliative’ as it would depend on context, so more appropriate peer-reviewed literature should be cited here.

Line 336-339: This entirely depends on the assumption that strains do not differ in their use of and sensitivity to the same compounds. This does not appear to be discussed, but an example is given later where this does not appear to be the case. Lines 445-450 indicates that darcin is expressed more in mice of the C57 lineage, but I don’t think ref 60 looks at SJL or AJ mice. A paper by Cheetham in 2009 (Physiology & Behavior) looks at lineage differences in expression of this protein, but this doesn’t correspond to the hypothesis here since the protein is not produced in both other low and high aggression strains (including SJL and AJ).

Line 497-499 Conclusion: It is hard to support this statement based on the current analysis. This really needs a much clearer demonstration of the alignment of strain specific VOC patterns and strain specific differences in the specific behaviors of interest.

7. PLOS authors have the option to publish the peer review history of their article (what does this mean?). If published, this will include your full peer review and any attached files.

Reviewer #1: No

Reviewer #2: No

---

## [Author Response · Author response to Decision Letter 1]

19 Mar 2021

All answers have been uploaded in a separate file, but are copied here:

We would like to start by saying thank you to the reviewers for providing such valuable commentary. Particularly, it is clear reviewer 2 spent a lot of time critiquing this manuscript and their insight truly contributed to strengthening this work and making it more comprehensible. We send our utmost gratitude for this effort. 

Additionally, we have updated the manuscript with a verified structure for MW 154 compound detected in sweat samples. 

Reviewer #1

I thank and congratulate the authors for their work to improve this manuscript. This report is very interesting and I am looking forward to reading up on future developments on the question of mouse aggression and plantar sweat. I would however suggest not to forget other odorant sources (e.g. urine or saliva), volatile compounds evaporate and can evade detection, and rarely biologically work in isolation (ie they are naturally occurring as part of mixtures, even though the literature glorifies single molecules like PEA, TMT and the likes).

Comments on ‘clean version’,

- Line 594 : is there a typo in this sentence?

This has been reworded

- Line 635: were utilized?

This has been corrected

- Throughout: I am not sure p’s is the plural of p, yet it could be this inappropriate use of apostrophe might be the best abbreviation for ‘p-values’

We have corrected “p’s” to “p values”. 

Reviewer #2 

I am happy that the authors have responded appropriately to my major concern that strain was not included in correlation analyses between VOCs from different sources and behaviors, and that sequential Bonferroni has now been applied to adjust p values for multiple comparisons. An oversight to include day where appropriate as a within-subjects factor to avoid pseudoreplication has been corrected. I also found that inclusion of the new Tables 3 and 4 was very helpful in showing the patterns found in the data analysis much more clearly, while new Figure 6 provides a useful summary from such a complex data analysis (though I didn’t understand the color coding, see also other comments on the figure below). My review therefore focuses on interpretation of the data analysis and conclusions drawn. Some specific suggestions for some rewording are provided at the end of this review.

The color scheme has been explained in Figure 6. 

The focus of the study is on identifying candidate VOCs in nest material that might account for reduced aggression observed previously when soiled nest material is transferred between cages at cage cleaning. VOCs associated with affiliative behavior (allogrooming and group sleep) are also examined, although relevance to the reduction in aggression in response to soiled nest material isn’t very clear, given aggression and affiliative behavior do not show a simple negative relationship (as confirmed in this study). As only affiliative but not aggressive behavior features in the title and much of the conclusions, this needs some further explanation as I struggled to understand the relevance to the main issue addressed here.

You are correct that altering aggression was our focus, but we did not know what to expect in terms of behavior diversity in the cage. Affiliative behaviors were included since previous work on laboratory mouse social behavior primarily does not take place in the home cage and little is published on what influences these behaviors in the laboratory and how that would relate to aggression. Thus, we did not want to risk overlooking social differences that may have been something other than agonistic behavior. Additionally, both promoting affiliative behavior and reducing aggression would benefit mouse welfare. This explanation has been added to the intro (line 70-76). 

The study is designed to look for VOC candidates that might have common effects in regulating aggression across mouse strains. I think it is important to clarify this underlying assumption, as differences between chosen strains in the odors used to regulate aggression, or in sensitivity to these VOCs, would result in a lack of association using this design. An example is mentioned in the Discussion (though not in this context) where strains differ in expression of a scent protein that stimulates aggression (lines 445-450), but nonetheless strain differences in expression do not correspond to differences in aggression but to strain lineage, which deserves further comment.

This assumption has been acknowledged in the end of the introduction and more explanation has been added to the discussion (line 106-107; 450-452 ).

Although not anticipated when selecting the three strains used, the study found that aggression within two of the strains was rare during the week-long study and common only in one of the strains (SJL). This is important information for interpretation of the findings but this only becomes apparent to the reader quite late in the Results section, from Figure 5. PCA analysis of behavior derived two components that distinguish the main strain differences, but it is important to refer to the data in Figure 5 to interpret these components given the major strain differences – currently the analysis is quite misleading (though I’m sure this is not intentionally so). Scores on PC1 (referred to as Aggression PC) are influenced not only by aggression and social investigation but also by high negative weighting to allo-grooming (not shaded in Table 2 but this has just as strong a weighting as others that are shaded). The highly significant difference between strains reported for this component (SJL > AJ > B6) is not because AJ has greater aggression than B6, which the text implies (Table 3, lines 191-192). Instead, the weightings on this component reflect greater aggression and social investigation in SJL compared to AJ and B6, but greater allogrooming in B6 than AJ or SJL (see Figure 5, leading to SJL > AJ > B6). This component thus discriminates the major differences in behavior between strains, but cannot be interpreted as reflecting aggression alone and the label Aggression PC is quite misleading. PC1 does not entirely explain behavioral differences between strains because group sleep shows a slightly different pattern (but accounts for less variance). The second component (referred to as Affiliative PC) reflects this strain difference in group sleep while being orthogonal to PC1 (which has already accounted for strain differences in aggression, investigation and allogrooming). Thus PC2 weightings reflect greater group sleep in B6 and SJL vs AJ while AJ also has lower allogrooming than B6 and lower aggression and social investigation than SJL so scores very low for this component. PCA merely finds the main axes of variation in the data – interpretation of the derived components and strain differences in scores requires inspection of the underlying behaviors shown in Figure 5, so it does not make sense to me to split up the analysis of behavior using PCA and the later section on strain and behavior.

Once the pattern of differences between strains is clear, one can then relate this in an appropriate way to VOC differences. As it stands, the implication of Table 3 and text is that aggression follows the pattern SJL > AJ > B6, so VOCs matching differences in aggression should be either SJL > AJ > B6 or SJL < AJ < B6. Instead, it is clear from Figure 5 that VOCs matching differences in aggression between strains should discriminate SJL from the other two strains and that “Aggression PC” is misleading.

Thank you for these comments and we agree with your suggestion about moving the sections. The behavior models have been moved earlier in the results section (line 209). Regarding our original approach to the behavior PCA, we had used contribution criteria to select meaningful variables, which in Behavior PC1, excluded allo-grooming. After your comment, we realized that we neglected to include this criteria in the manuscript, but also, upon further investigation, we determined this criteria is more suitable for identifying influential observations/sample units, not variables. Therefore allo-grooming has been included in the behavior PC1 interpretation and have removed our “Aggression” and “Affiliative” labels to avoid confusion. 

With that in mind, nest PC1 scores differ between SJL and the other two strains. However, as data on levels of the main compounds that contribute to nest PC1 are not presented, we cannot see which components differ consistently between SJL and both other strains (consistent with the hypothesis that some VOC components match strain differences in aggression) – important as the difference here is only just statistically significant while the strain difference in behavior is substantial. Would it not be better to focus detailed descriptions on this specific nest PC1 component as the one that does correspond with the strain difference of interest while also relating to nest material known to have aggression reducing properties, rather than losing the key relationship here in all the detail of VOC profiles that do not match the hypothesis being tested? The behavioral analysis clearly shows that the pattern of neither allogrooming nor group sleep (referred to as affiliative behaviors) correspond to strain differences in aggression, which needs to be pointed out much more clearly since the point of discussing VOC patterns in relation to ‘affiliative behavior’ (which do not correlate across strains and cannot simply be grouped together) is very unclear.

I do think there are interesting and novel data here, but this is being lost in an over-complex analysis that seems to be leading to confusion rather than insight. I would really advise the authors to consider focusing first on the strain differences in behavior which are quite clear cut. The finding that SJL was the only strain to show appreciable aggression then changes the focus of analysis of VOC profiles and could be presented in a very straightforward way. Allogrooming and group sleep each show different strain patterns so it is not possible to simply group these together as ‘affiliative behaviors’ in relation to strain differences from these analyses, and authors need to avoid the implication that mice that are affiliative are therefore not aggressive. If the main focus of the study is on VOCs that might appease aggression, shouldn’t the title reflect the study findings in relation to aggression not in relation to uncorrelated affiliative behaviors? It is just a suggestion, but throughout I found this confusing.

The discussion is extremely long and interpretations throughout really need revisiting in light of the points made above concerning the interpretation of behavior PCs and their match to VOCs. As a suggestion, perhaps some of the more speculative aspects could be dropped so that the focus remains clearly on the findings of the study.

Thank you for these suggestions. The previous behavior PC labels have been removed and the discussion has been streamlined. We fully admit that we were so focused on all of the details from these data that we lost track of our main message. Thank you for helping us see that as well as help focus our manuscript. The discussion now focuses specifically on nest PC1 and urine PC2 (since their strain patterns match that of observed behavior) and sweat PC1 (since sweat was central to our hypothesis). 

Some specific points, particularly in relation to conclusions from analyses

Abstract line 33: The differences according to strain do not ‘mask direct effects on behavior’ in these analyses. Including strain in analyses increased the likelihood of seeing overall correlations between individual VOC profiles and behaviors, by taking into account any differences in levels of behavior or VOC profiles between strains. Instead analyses showed that there were no consistent relationships between individual VOC profiles and behavior that were not accounted for by major strain differences.

This line has been reworded.

Abstract lines 34-37: As the data show that the two affiliative behaviors analysed were not correlated (thus would be better referred to separately), while differences in these behaviors between strains did not match differences in aggression, this seems to miss the main results and conclusions that can be drawn from the data.

These lines have been reworded.

Abstract line 37: The term ‘correlation’ is not appropriate here as this implies a continuous statistical relationship which was not shown with 3 strains. Strain differences in VOC patterns that match strain differences in behavior would be more appropriate as a conclusion, being appropriately cautious here as there is no statistical validity in just picking out the VOC profiles that match.

This line has been reworded.

Abstract line 42: ‘provide preliminary information about a potential connection between …..

Several changes to the abstract were made however we are unsure what alteration the reviewer was requesting here. 

Intro line 66-68: As affiliative behavior is often not negatively correlated with aggression (more social animals generally show more of both), the relevance of affiliative behavior to the issue of appeasing odors in nest material needs more explanation. Does addition of soiled nest material result in increased affiliative behaviors as well as reduced aggression?

More explanation on why affiliative behaviors are of interest has been added (line 70-76).

Intro line 81: This sentence doesn’t seem to make sense. Do you mean the mechanism behind the reduction in aggression observed in response to used nest material?

This line has been reworded.

Intro lines 91-104: There needs to be some introduction to the choice of strains included in the study here, particularly when the next section is Results rather than Methods. Readers may not be familiar with these strains and what is already known about their aggressive / affiliative behavior.

A brief strain overview has been added to the intro (line 102-104)

Intro, last sentence: This would be more suitable for Discussion rather than introduction to this study.

This line has been moved. 

Results (specific comments, see main comments above about strain differences and interpretation of PCA components)

Line 128: I would have found ‘Sample VOC profiles’ more helpful here (everything is a sample)

This line has been reworded

Line 133-134: I couldn’t see any indication in Table 1 for which compounds had been previously identified as mouse urinary compounds.

An additional symbol has been added to denote urinary signals.

Line 173: I would avoid the term ‘significant’ here as it implies statistical significance relative to random.

This line has been reworded.

Line 176: See main comments – I think giving these interpretive terms to the behavior PCs is misleading given the actual strain differences in the behaviors contributing to these components.

These labels have been removed. 

Line 178: Again, there is no validity to using ‘significant’ here perhaps ‘considered important’. As indicated above, this really requires inspection of the individual component data for interpretation but this hasn’t been included.

This line has been reworded.

Table 2: ‘mediated’ and ‘escalated’ mean nothing to the reader – there is space in the Behavior column to include ‘aggression’.

These labels have been clarified. 

Table 4: Given the importance of strain differences in these different behaviors for any interpretation, could you indicate in the table which strains are significantly different from which (as in Table 3? As none of the strain x day interactions are statistically significant, this could be simply stated in a footnote, giving appropriate space for showing the pattern of strain differences. As indicated in main comments, I think this section needs to come much earlier so that the PCA analyses and behavioral match to VOC patterns are interpretable. Given that aggression was so much stronger in SJL than in the other two strains, it would make more sense to match VOC patterns to this difference than to principal components that represent a much more complex pattern of differences between strains.

This table has been restructured and this section was moved (line 209-236). Thank you, this suggestion greatly improves the flow of the paper. 

Line 277: In summary, SJL displayed substantially more aggressive behavior?

This line has been reworded. 

Line 279 & Figure 6: B6 mice did not display more group sleep than SJL

This line has been reworded and group sleep was removed from the figure. As mentioned above, the color coding has been explained in the Figure 6 legend.

Discussion

Line 292: As indicated, Aggression PC does not only include aggression, but your data do show that only SJL showed appreciable aggression.

This line has been edited to refer to aggression directly. 

Line 299: Some of the component weights in nest PC1 are negative (low levels would result in high scores)

Negative weights on nest PC1 have been included in this line. 

Lines 297- 301, 315-330: Does the pattern of urine PC2 scores really match the differences in aggression (not shown – you could plot this)?

Measures of correlation have been added between urine PC2 and aggression, as well as nest PC1 and aggression. (line 272-274)

Line 334-335: Surely sniffing behavior denotes gaining olfactory information. I don’t think anyone working in olfactory communication would define this as ‘affiliative’ as it would depend on context, so more appropriate peer-reviewed literature should be cited here.

The affiliative classification has been removed from this line. We wanted to be more general in regards to how sniffing is classified for mouse behavior. We don’t want to speculate further since a more in depth evaluation of different types of sniffing (snout/urogenital) and their motivation was not conducted.

Line 336-339: This entirely depends on the assumption that strains do not differ in their use of and sensitivity to the same compounds. This does not appear to be discussed, but an example is given later where this does not appear to be the case. Lines 445-450 indicates that darcin is expressed more in mice of the C57 lineage, but I don’t think ref 60 looks at SJL or AJ mice. A paper by Cheetham in 2009 (Physiology & Behavior) looks at lineage differences in expression of this protein, but this doesn’t correspond to the hypothesis here since the protein is not produced in both other low and high aggression strains (including SJL and AJ).

The darcin example was about strain specific expression, but it does not imply that sensitivity depends on expression. A reference to Kaur et al. 2014 has been added to this section. Admittedly, it uses BALB/c mice, but they are of the same lineage as AJ. Based on our literature searches, pheromone sensitivity has not been assessed in AJ or SJL mice. 

Line 497-499 Conclusion: It is hard to support this statement based on the current analysis. This really needs a much clearer demonstration of the alignment of strain specific VOC patterns and strain specific differences in the specific behaviors of interest.

Additional analyses of high loading compounds on nest PC1, urine PC2, and sweat PC1 have been provided to justify this statement. (line 250-278; 282 (Table 5))

---

## [Decision Letter · Decision Letter 2]

27 Apr 2021

Compounds from plantar foot sweat, nesting material, and urine show strain patterns associated with agonistic and affiliative behaviors in group housed male mice, Mus musculus

PONE-D-20-24539R2

Dear Dr. Barabas,

We’re pleased to inform you that your manuscript has been judged scientifically suitable for publication and will be formally accepted for publication once it meets all outstanding technical requirements.

Kind regards,

Igor Branchi, Ph.D.

Academic Editor

PLOS ONE

Additional Editor Comments (optional):

Reviewers' comments:

Reviewer's Responses to Questions

**Comments to the Author**

1. If the authors have adequately addressed your comments raised in a previous round of review and you feel that this manuscript is now acceptable for publication, you may indicate that here to bypass the “Comments to the Author” section, enter your conflict of interest statement in the “Confidential to Editor” section, and submit your "Accept" recommendation.

Reviewer #1: All comments have been addressed

Reviewer #2: (No Response)

2. Is the manuscript technically sound, and do the data support the conclusions?

Reviewer #1: Yes

Reviewer #2: Yes

3. Has the statistical analysis been performed appropriately and rigorously? 

Reviewer #1: I Don't Know

Reviewer #2: Yes

4. Have the authors made all data underlying the findings in their manuscript fully available?

Reviewer #1: Yes

Reviewer #2: Yes

5. Is the manuscript presented in an intelligible fashion and written in standard English?

Reviewer #1: Yes

Reviewer #2: Yes

6. Review Comments to the Author

Reviewer #1: Apologies for the delay in providing this review. Thanks to the statistical skills and hard work of reviewer 2, who raised issues I had not appreciated, Barrabas et al resubmitted the manuscript with updated title, additional author, updated analysis and more data (incl the identity of compound MW 154).

My failure to raise those data analysis points in the first place was due to my partial understanding of the statistical processes, and therefore, although the comments/responses/updates I have read appear to make sense to me, I am not able to fully appraise the new analysis.

At this stage there is a risk for this reviewer of reading the manuscript with fresh eyes, and finding new issues I had not brought up before. So I will try to keep this brief and will leave it to the authors and editors to decide. None of these points should prevent publication of this very interesting article.

Line 53 (and line 89): I was curious about the data behind the practice of transferring used nest material to a new cage as a way to reduce mouse aggression (here it’s ref 12). For instance, here is one of the concluding sentences of reference 10 (Gray, S., Hurst, J.L., 1995): ‘We would thus recommend that mice are transferred into completely clean cages when aggression within caged groups of males is a concern’. Could the authors briefly discuss that there does not appear to be a full consensus in the literature? The only reason for doing this is to avoid the propagation of myths and stimulate debates.

Line 340-343: how is this not speculative anymore?

Line 359 : correct ininafter

Line 362: is this backed by a reference or anecdotal evidence?

Line 389-390: I am not sure I understand the sentence

Assuming no further versions of this work are to be reviewed, I would like to thank the authors and reviewer 2, this has been a very interesting and edifying process.

Reviewer #2: I think the authors have done a great job in responding to my comments and suggestions, this paper makes a very valuable contribution to literature in this area.

7. PLOS authors have the option to publish the peer review history of their article (what does this mean?). If published, this will include your full peer review and any attached files.

Reviewer #1: No

Reviewer #2: **Yes: **Jane L. Hurst

---

## [Editor Report · Acceptance letter]

5 May 2021

PONE-D-20-24539R2 

Compounds from plantar foot sweat, nesting material, and urine show strain patterns associated with agonistic and affiliative behaviors in group housed male mice, *Mus musculus*

Dear Dr. Barabas:

I'm pleased to inform you that your manuscript has been deemed suitable for publication in PLOS ONE. Congratulations! Your manuscript is now with our production department. 

Kind regards, 

on behalf of

Dr. Igor Branchi 

Academic Editor

PLOS ONE